# A refined method for calculating Equivalent Effective Stratospheric Chlorine.

Andreas Engel, Harald Bönisch[1], Jennifer Ostermöller, Sandip Dhomse[2], Martyn P. Chipperfield[2] And Patrick Jöckel[3].

University of Frankfurt, Institute for Atmospheric and Environmental Sciences, Altenhöferallee 1, 60438 Frankfurt, Germany
[1]now at Karlsruhe Institute of Technology, Institute of Meteorology and Climate Research, Hermann-von-Helmholtz-Platz 1, 76344 Eggenstein-Leopoldshafen, Germany.
[2]Institute for Climate and Atmospheric Science, School of Earth and Environment, University of Leeds, Leeds, LS2 9JT, U.K.
[3]Deutsches Zentrum für Luft- und Raumfahrt (DLR), Institut für Physik der Atmosphäre, Oberpfaffenhofen, Germany.

*Correspondence to*: Andreas Engel (an.engel@iau.uni-frankfurt.de)

**Abstract.** Chlorine and bromine atoms lead to catalytic depletion of ozone in the stratosphere. Therefore the use and production of ozone depleting-substances (ODS) containing chlorine and bromine is regulated by the Montreal Protocol to protect the ozone layer. Equivalent Effective Stratospheric Chlorine (EESC) has been adopted as an appropriate metric to describe the combined effects of chlorine and bromine released from halocarbons on stratospheric ozone. Here we revisit the concept of calculating EESC. We derive a refined formulation of EESC based on an advanced concept of ODS propagation into the stratosphere and reactive halogen release. A new transit time distribution is introduced in which the age spectrum for an inert tracer is weighted with the release function for inorganic halogen from the source gases. This distribution is termed the "release time distribution". We show that a much better agreement to inorganic halogen loading from the chemistry transport model TOMCAT is achieved than using the current formulation. The refined formulation shows EESC levels in the year 1980 for the mid latitude lower stratosphere, which are significantly lower than previously calculated. 1980 marks the year commonly used as a benchmark to which EESC must return in order to reach significant progress towards halogen and ozone recovery. Assuming that – under otherwise unchanged conditions - the EESC value must return to the same level in order for ozone to fully recover, we show that it will take more than 10 years longer than estimated in this region of the stratosphere with the current method for calculation of EESC. We also present a range of sensitivity studies to investigate the effect of changes and uncertainties in the fractional release factors and in the assumptions on the shape of the release time distributions. We further discuss the value of EESC as a proxy for future evolution of inorganic halogen loading under changing atmospheric dynamics using simulations from the EMAC model. We show that while the expected changes in stratospheric transport lead to significant differences between EESC and modelled inorganic halogen loading at constant mean age, EESC is a reasonable proxy for modelled inorganic halogen on a constant pressure level.

# 1 Introduction

It is well established that chlorine and bromine atoms in the stratosphere enhance ozone loss via catalytic reaction chains (Stolarski. and Cicerone, 1974;Solomon, 1999;Molina and Rowland, 1974;Wofsy et al., 1975). Ozone depletion has been observed at mid latitudes (S. Pawson and W. Steinbrecht et al., 2014) and in particular at high latitudes during winter and spring time (Farman et al., 1985;M. Dameris and S. Godin-Beekmann et al., 2014). The chlorine and bromine atoms responsible for the ozone depletion are not injected directly into the stratosphere, but are released from organic halocarbons, so called ozone depleting substances (ODS), which are emitted in the troposphere. Ozone is thus not depleted by reactions with the chemicals emitted, but by reaction with the inorganic halogen released from these chemicals. The extent of the catalytic ozone depletion depends on the amount of inorganic halogen in the stratosphere. In models which include both chemistry and transport of the stratosphere, the amount of inorganic halogen can be directly calculated as $Cl_y$ and $Br_y$. On average, the inorganic halogen content is larger for air parcels with higher mean age (Newman et al., 2007;Engel et al., 1997), but the relation between $Cl_y$ and mean age may differ from model to model depending on the representation of transport and chemistry in the model (Waugh et al., 2007). The relation between mean age and $Cl_y$ depends on the interaction between transport and chemistry and is a function of both, the time spent in the stratosphere, and the transport pathways (Hall, 2000;Schoeberl et al., 2000;Schoeberl et al., 2005;Waugh et al., 2007). Equivalent Effective Stratospheric Chlorine (EESC) is a metric describing the combined effect of all chlorinated and brominated ODSs expressed as the equivalent amount of inorganic chlorine in the stratosphere based on tropospheric abundances of tropospheric source gases (Daniel and Velders, 2011;Newman et al., 2007). While in principle, $N_2O$ could be included in EESC, as the $NO_x$ released in the stratosphere also leads to ozone depletion (Ravishankara et al., 2009), this is complicated, as the efficiency with which $N_2O$ leads to ozone depletion changes with halogen loading (Daniel et al., 2010;Ravishankara et al., 2009). We therefore do not include $N_2O$ in EESC. EESC depends on the transport from the troposphere into the stratosphere, the temporal trend of the mixing ratios of the source gases in the troposphere and the release of inorganic halogen from these source gases. EESC has been used widely as a proxy to describe the combined effects of inorganic bromine and chlorine on stratospheric ozone, e.g. in the analysis of time series of ozone or when discussing the effects of volcanoes or geoengineering (Tilmes et al., 2009;Shepherd et al., 2014;Weatherhead and Andersen, 2006;Chipperfield et al., 2017). Note that EESC is only a valid proxy for anthropogenic ozone depletion, if all other parameters, especially atmospheric transport, are unchanged. A projection of a return of EESC to some specific level therefore does not imply that ozone will return to the same levels. EESC should thus be regarded as proxy for the impact of halogenated source gases on the ozone layer due to both anthropogenic and natural emissions. The recovery of the ozone layer is affected by other parameters in addition, especially changes in transport and temperature. EESC is therefore not a proxy for ozone recovery, but a proxy for the impact due to one single parameter, the halogen loading. In section 4.3. we discuss the validity of EESC as a proxy for inorganic halogen loading under the influence of changing stratospheric dynamics.

The transport into and within the stratosphere is described by the mean age of air, Γ (Hall and Plumb, 1994;Waugh and Hall, 2002;Kida, 1983). A stratospheric air parcel does not have a single transit time $t'$ since its entry into the stratosphere, but is rather composed of a large number of irreversibly mixed fragments or fluid elements with varying transit times $t'$, describing the variable times they already spent in the stratosphere. The distribution of transit times is called the age spectrum, the arithmetic mean (first moment) being the mean age Γ. The age spectrum is generally described by a Green's function $G$ for one-dimensional advective diffusive transport and a parameterization of the width of the distribution as a function of the mean age (Hall and Plumb, 1994). Together with the temporal trend of the trace gas in the troposphere, the age spectrum determines the mixing ratio of an inert trace gas in the stratosphere (1) at a certain time $t$ and place $r$, $\chi_{inert,strat}(r,t)$, as the fluid elements will each contain the mixing ratio present in the troposphere at the time they entered the stratosphere, $\chi_0(t - t')$. Based on this concept, it is also possible to derive mean age of air (Hall and Plumb, 1994;Volk et al., 1997;Engel et al., 2002;Engel et al., 2009) based on observations of chemically inert tracers in the stratosphere, so called age tracer.

$$\chi_{inert,strat}(r,t) = \int_0^\infty \chi_0(t - t') \cdot G(r,t')dt' \tag{1}$$

For chemically active species, in addition to the transport, the chemical loss leading to the release of inorganic halogen needs to be considered. This release of inorganic chlorine and bromine from the halocarbon source gases is characterized by the fractional release factor $f$ (FRF). The FRF describes which fraction of the source gas molecules originally present in an air parcel has already been released, i.e. transferred to the inorganic fraction. $1-f$ will thus describe the fraction that is still in the form of the organic source gas. Fractional release factors for many relevant trace gases have been determined as a function of mean age (Newman et al., 2007;Laube et al., 2013). Typically, a mean age value of 3 years is adapted for the lower stratosphere of the mid latitudes and a mean age value of 5.5 years is used for polar winter conditions in the lower stratosphere (Newman et al., 2007). The calculation of $f$ relies on the difference of the observed mixing ratio of the source gases in the stratosphere to the amount of source gas originally present in this air-parcel (2). For this, a reference mixing $\chi_{ref}(\Gamma)$ ratio must be determined, based on temporal trends in the troposphere and transport into the stratosphere.

$$f(\Gamma) = \frac{\chi_{ref}(\Gamma) - \chi_{strat}(\Gamma)}{\chi_{ref}(\Gamma)} \tag{2}$$

This reference mixing ratio has typically been calculated using (1), i.e. assuming that the chemically active gas propagates in the same way as a chemically inert gas. Plumb et al. (1999) showed that the age spectrum $G$, which is representative for an inert gas is not well suited to describe the way that a chemical active gas is propagated into the stratosphere. The reason for this is that the remaining organic fraction of a chemically active species (CAS) is determined largely by the fluid elements with shorter transit times, where chemical loss is less pronounced. The fluid elements with longer transit times on the other hand do not contribute as much to the remaining organic fraction, as more chemical loss has occurred. The combination of chemical loss and transport is described by a modified age spectrum, called the arrival time distribution. This arrival time

distribution is weighted stronger at shorter transit times and has a different first moment than the age spectrum for an inert tracer. This first moment of the arrival time distribution has a lower value than the mean age and is termed the mean arrival time. Information on the mean arrival time was derived from 2-D model calculations by Plumb et al. (1999), who used the mean arrival time to detrend stratospheric correlations. They found that the detrended correlations from different years showed good agreement, if the mean arrival time was used in calculating the reference values, while this was not the case when using mean age. Ostermöller et al. (2017) could show that the arrival time distribution also allows to derive fractional release factors, which are not influenced by the tropospheric trend. These studies show that, due to the interaction of chemistry and transport, changes in the tropospheric mixing ratios of source gases with chemical loss are reflected faster in their stratospheric mixing ratios than are changes in gases without chemical loss. The age spectrum for in inert tracer is not well suited to describe the propagation of a tracer with chemical loss into the stratosphere.

EESC is influenced by the temporal trends of the source gases, their fractional release factors and the transport into the stratosphere. While all these factors may vary with time, e.g. due to changes in stratospheric circulation (Douglass et al., 2008;Li et al., 2012b), especially fractional release factors should not depend on the tropospheric trends of the trace gases (Ostermöller et al., 2017). As in the case of FRF, EESC is usually calculated as a function of mean age and again a mean age value of 3 years is adapted for the lower stratosphere of the middle latitudes and a mean age value of 5.5 years is used for polar winter conditions in the lower stratosphere (Newman et al., 2007). The formulation which is currently used to calculate EESC is based on the concept of fractional release and mean age, using the age spectrum $G$ for an inert tracer. In this formulation EESC is calculated by multiplying the fractional release factor with the integral over the tropospheric time series $\chi_0$ of the tracer and the age spectrum for an inert tracer (Newman et al., 2007;Velders and Daniel, 2014)

$$EESC_{current}(\Gamma, t) = \sum_{Cl}\left( n_i f_i(\Gamma) \int_0^\infty \chi_{0,i}(t - t')G(\Gamma, t')dt' \right) + \alpha \sum_{Br}\left( n_i f_i(\Gamma) \int_0^\infty \chi_{0,i}(t - t')G(\Gamma, t')dt' \right) \tag{3}$$

with $n_i$ being the number of chlorine or bromine atoms in species $i$ and $f_i$ being the fractional release factor. $\alpha$ is a factor representing the higher effectivity of bromine to ozone depletion, typically taken as 60 for both high latitudes and mid latitudes (Newman et al., 2007). The age spectrum $G$ used here is that for an inert tracer. As shown by Plumb et al. (1999) and Ostermöller et al. (2017), the arrival time distribution is better suited to describe the propagation of the organic fraction of a source gas into the stratosphere. Consequently, it is also expected that the age spectrum for an inert tracer may not be the best way to describe the propagation and release of the inorganic fraction and thus EESC.

In this paper we discuss the interaction of chemistry and transport in the propagation of chemically active tracers with tropospheric trends into the stratosphere and suggest an improved method for the calculation of EESC. The paper is organized as follows. In section 2 we present some general thoughts on the propagation of tropospheric trends taking into account

chemical loss. In section 3 we derive a new mathematical formulation for EESC, based on the ideas developed in section 2. This new mathematical formulation is applied in section 4 to the scenario of source gas mixing ratios given by Velders and Daniel (2014) and the results are compared to their results for the estimated recovery of EESC to 1980 values. We also present sensitivity studies of the new EESC formulation to different parameters and compare the different formulations of EESC to simulations of inorganic halogen loading from two different comprehensive three-dimensional atmospheric chemistry models. Finally we draw some conclusion and present an outlook in section 5.

## 2. On the influence of transport and tropospheric trends on chemically active species.

In addition to transport and temporal trends in the troposphere, the stratospheric mixing ratio of a species with chemical loss in the stratosphere depends on the loss processes and on the interplay between transport, chemical loss and the temporal trend (Volk et al., 1997;Plumb et al., 1999). The age spectrum $G$, which is used to describe the propagation of chemically inert trace gases into the stratosphere (Schauffler et al., 2003;Newman et al., 2007;Engel et al., 2002) and to calculate mean age does not take into account chemical loss. Chemical loss is not uniform throughout the stratosphere, as it depends in most cases on the actinic flux at short wavelength. In the annual mean, the local chemical lifetime will in general decrease with altitude and increase with increasing latitude, leading to a clear relation between the maximum altitude of a fluid element and the fractional release (Douglass et al., 2008;Hall, 2000). The chemical loss is thus very inhomogeneous, but on average, the longer a fluid element remains in the stratosphere, the larger the integrated chemical loss will be (Plumb et al., 1999). Also, as the loss for most species with photochemical sinks mainly occurs at higher altitudes and tropical latitudes in the stratosphere, it is expected that, on average, longer transit times will be associated with shorter local lifetimes along the transport pathway. A transit time distribution in which the transit times are weighted with the transit time dependent chemical loss has been termed "arrival time distribution" (Plumb et al., 1999), $G^*$ (4). The fractional chemical loss can be expressed in a very generalized way as $(1 - f(t', p))$, where $f(t', p)$ is a fractional release function, which is specific for each trace gas and will depend on the time the air parcel has spent in the stratosphere t' and on the path p it has take, primarily the maximum path height (MPH) during transport (Hall, 2000). While the path or the MPH are not known, it has been shown that "molecules arriving at X with long arrival times will, on average, have spent more time exposed to chemical loss and will have sampled atmospheric regions where photochemical loss is greater" (Plumb et al., 1999). We therefore follow the approach that the fractional loss can on average be described as function of the transit time only. That chemical loss and transit time are on the average related to each other is also reflected in the tight observed correlations between mean age and tracer mixing ratios (e.g.Volk et al., 1997;Engel et al., 2002). While there will be fluid elements with very different paths and different chemical loss which have the same transit time the loss can on average be sufficiently well described as a function of the transit time. We therefore treat $f(t', p)$ as $f(t')$ only. This is also in line with the findings of Schoeberl et al. (2000), who showed that using an "average path approximation" with a "single-path photochemistry" and thus with a unique relationship between loss and transit time, global tracer-tracer correlations can be explained. This concept that loss can be described only as a function of transit time without

considering the different transit pathways was also adopted by Schoeberl et al. (2005) in the derivation of age spectra. We use a mean age of 3 years for mid latitudes and of 5.5. years for high latitudes. Indeed, the path distribution for an air parcel with mean age of 3 years in the tropics, in mid latitudes and in polar regions is expected to show more variability than for air parcels investigated under similar conditions (e.g.latitude regions). As this analysis is restricted to one latitude band for one mean age

level, we therefore approximate loss as a function of transit time only. The first moment of the arrival time distribution is called the mean arrival time $\Gamma^*$. This distribution describes the probability distribution for organic source gas molecules to arrive at some place $r$ in the stratosphere, as a function of transit time $t'$

$$G^*(r,t') \equiv (1 - f(t')) \cdot G(r,t'). \tag{4}$$

We can now define a second transit time distribution, $G^\#(r,t')$, which describes the probability for an inorganic halogen atom released from this source gas to arrive at this place $r$ in the stratosphere, again as a function of transit time $t'$

$$G^\#(r,t') \equiv f(t') \cdot G(r,t'). \tag{5}$$

For the calculation of $G^*$ and $G^\#$ the integrated loss as a function of transit time needs to be known. Purely for illustrative purposes, we have constructed such a loss function using a sigmoid function, which changes from 0 (no loss) for short transit times to 1 (complete loss) for longer transit times. The function has been constructed in a way to match fractional release factors for CFC-11 for 3 and 5.5. years. CFC-11 is one of the most important chlorine source gases for the stratosphere. These three different transit time distributions, $G$, $G^*$ and $G^\#$ calculated using typical age spectra for 3 and 5.5 years of mean age are

shown in Figure 1 and Figure 2. In both cases, the mean transit time for the inorganic fraction is longer than that for an inert tracer, while the lag of the remaining organic fraction is shorter than that of an inert tracer. The effect is much more pronounced for the 3 years mean age calculation, where the fractional release is about 0.5. i.e. the organic and the inorganic fraction are about equal. In the case of 5.5 years mean age, nearly all CFC-11 molecules are converted to the inorganic form and the remaining organic fraction gets very small. The mean transit times for all three distributions are calculated as the arithmetic

mean or first moment of the respective distribution functions. The mean transit time of the organic fraction is described by the mean arrival time $\Gamma^*$ (Plumb et al., 1999), that of an inert tracer by the mean age $\Gamma$ . The inorganic fraction is described by a third time scale, which represents a release weighted transit time distribution. We suggest the term "release time distribution", $G^\#$, for this transit time distribution with a first moment called the "mean release time", $\Gamma^\#$. In the example given in Figure 1 the mean age $\Gamma$ is 3 years, the mean arrival time $\Gamma^*$ for the organic fraction is 1.75 years and the mean release time $\Gamma^\#$

describing the inorganic fraction is 4.35 years. Inorganic chlorine thus lags the tropospheric time series more than expected from an inert tracer. Mean arrival time $\Gamma^*$ and mean release time $\Gamma^\#$ differ for each tracer depending on their chemical loss behavior, which is described by the fractional release factor $f$. A parameterization of the mean arrival time $\Gamma^*$ for all the relevant chlorine and bromine species has been calculated as a function of lifetime and mean age (Plumb et al., 1999). The mean release time $\Gamma^\#$ can be derived from $\Gamma$, $\Gamma^*$ and the fractional release factor (see section 3.1). As EESC is a proxy for inorganic halogen,

we derive a new formulation of EESC which takes into account this interaction between chemistry and transport in an improved way.

## 3 Deriving a new formulation of EESC

The new mathematical formulation of EESC proposed here is derived based on the concept of how a trace gas of tropospheric origin with a temporal trend and chemical loss in the stratosphere propagates into the stratosphere. The organic source gases of chlorine and bromine are such gases. In order to derive the amount of inorganic chlorine or bromine that has been released from such an organic source gas at some point $r$ in the stratosphere, three different functions must be considered, which are all functions of the transit time. We will denote transit time, i.e. the time a fluid element has spent in the stratosphere as $t'$, while the time itself will be denoted as $t$. First, the transit time distribution, i.e. how long it has taken for the individual fluid elements of this air parcel to travel from their entry point to the stratosphere to the location r in the stratosphere. Second, the temporal trend of the mixing ratios at the entry point has to be considered and, third, chemical loss during this transport. All three functions depend on the transit time $t'$. The integral over all possible transit times over these three functions will yield the remaining mixing ratio of the source gas. If we denote the time series at the entry point to the stratosphere as $\chi_0(t - t')$, the transit time distribution for air to reach some point $r$ in the stratosphere as $G(r, t')$ and the chemical loss term, which can be be described by the factor $\left(1 - f(t')\right)$, where $f(t')$ describes the fraction which has been lost.

$$\chi_{strat}(r, t) = \int_0^\infty \chi_0(t - t') \cdot \left(1 - f(t')\right) \cdot G(r, t')dt' \tag{6}$$

As $\left(1 - f(t')\right)$ is the remaining fraction of the organic source gas, the mixing ratio of inorganic chlorine released from the source gas would then be

$$\chi_{inorg,strat}(r, t) = \int_0^\infty \chi_0(t - t') \cdot f(t') \cdot G(r, t')dt' \tag{7}$$

For simplicity, we have assumed here that the source gas releases only one atom of inorganic halogen. Transport and mixing are described by the transit time distribution, also known as the age spectrum or Green's function $G$. $G$ describes the probability of a certain transit time since entry into the stratosphere at the tropical tropopause, and thus describes both net mass transport and mixing. $G$ is a function of transit time $t'$ and the location in the stratosphere, $r$. The integral over the probability of all transit times must be equal to 1.

$$\int_0^\infty G(r, t')dt' = 1 \tag{8}$$

and the integral over all transit times weighted by their probability is the mean age of air $\Gamma$ (Hall and Plumb, 1994).

$$\Gamma(r) = \int_0^\infty t' \cdot G(r,t')dt' \qquad (9)$$

We now use the new transit time distribution $G^\#$ that was introduced in section 2. $G^\#$ is defined as the product of the transit time dependent fractional release factor $f(t')$ and the age spectrum.

$$G^\#(r,t') \equiv f(t') \cdot G(r,t') \qquad (10)$$

As $G^\#$ is the product of the fractional release and the transit time distribution, it represents a release weighted transit time distribution. We will refer to this distribution as the release time distribution. Note that the integral over $G^\#$ is only unity in case of complete loss of the organic faction, i.e. $f(t')$ is 1 for all transit times $t'$. In all cases, the integral must be less or equal to 1.

$$\int_0^\infty G^\#(r,t')dt' \leq 1 \qquad (11).$$

We can, however, define a new, normalized release time distribution $G_N^\#$, by dividing $G^\#$ through the integral of $G^\#$ over all possible transit times

$$G_N^\#(r,t') \equiv \frac{G^\#(r,t')}{\int_0^\infty G^\#(r,t')dt'} = \frac{G^\#(r,t')}{\int_0^\infty f(t') \cdot G(r,t')dt'} \qquad (12).$$

The integral over $G_N^\#$ over all possible transit times is now unity

$$\int_0^\infty G_N^\#(r,t')dt' = 1. \qquad (13)$$

The integral over all transit times weighted by the normalized release time distribution $G_N^\#$ yields a "mean release time", $\Gamma^\#$, as also shown in Figures 1 and 2:

$$\int_0^\infty t' \cdot G_N^\#(r,t')dt' = \Gamma^\#. \qquad (14)$$

The integral in the denominator of (12) represents the first moment of the distribution of all fractional release factors, thus a mean fractional release factor, which is a function of the location $r$ in the stratosphere (Ostermöller et al., 2017), but in contrast to $f(t')$, it is not a function of transit time anymore:

$$\int_0^\infty G^\#(r,t')dt' = \int_0^\infty f(t') \cdot G(r,t')dt' = \overline{f}(r). \qquad (15)$$

Inserting (15) into (12) and solving for $G^\#$ yields:

$$G^\#(r,t') = G_N^\#(r,t') \cdot \overline{f}(r) \qquad (16)$$

Using the definition of $G^{\#}$ (5) we can thus derive a relationship between $G$ and $G^{\#}$

$$G^{\#}(r,t') = G_N^{\#}(r,t') \cdot \overline{f}(r) = G(r,t') \cdot f(t'). \tag{17}$$

The term $f(t') \cdot G(r,t')$ in (7) can thus be replaced by $G_N^{\#}(r,t') \cdot \overline{f}(r)$ to derive a new relationship for inorganic chlorine

$$\chi_{inorg,strat}(r,t) = \int_0^{\infty} \chi_0(t-t') \cdot \overline{f}(r) \cdot G_N^{\#}(r,t')dt'. \tag{18}$$

In contrast to $f(t')$, $\overline{f}(r)$ is independent of $t'$ and it can be extracted from the integral and (18) can be rewritten:

$$\chi_{inorg,strat}(r,t) = \overline{f}(r) \cdot \int_0^{\infty} \chi_0(t-t') \cdot G_N^{\#}(r,t')dt'. \tag{19}$$

Instead of describing the mixing ratio of inorganic chlorine at some location $r$ in the stratosphere, we can also describe it as a function of a certain mean age value $\Gamma$. This implies that at all locations r with the same mean release time, the release time distribution is the same. This assumption may not be valid everywhere, but as a mean age of 3 years is used for mid latitudes and of 5.5 years for high latitudes, we use this assumption only for air parcels under similar meteorological conditions (latitude bands). The release time distribution is then expressed as a function of mean release time $\Gamma^{\#}$. Equation (19) then becomes

$$\chi_{inorg,strat}(\Gamma,t) = \overline{f}(\Gamma) \cdot \int_0^{\infty} \chi_0(t-t') \cdot G_N^{\#}(\Gamma^{\#},t')dt'. \tag{20}$$

After multiplying the right hand side of (20) with the amount of halogen atoms released from a halocarbon ($n_i$) and in case of bromine with the factor $\alpha$ describing the relative efficiency of bromine and summing up over all halogen species $i$ we arrive at

$$EESC_{new}(\Gamma,t) = \sum_{Cl}\left( n_i\overline{f}_i(\Gamma) \int_0^{\infty} \chi_{0,i}(t-t')G_{N,i}^{\#}(\Gamma^{\#},t')dt' \right) + \alpha \sum_{Br}\left( n_i\overline{f}_i(\Gamma) \int_0^{\infty} \chi_{0,i}(t-t')G_{N,i}^{\#}(\Gamma^{\#},t')dt' \right) \tag{21},$$

which is the new formulation we suggest for the calculation of EESC. This formulation is similar to the one used by Velders and Daniel (2014) and Newman et al. (2007) and also in the most recent WMO ozone assessment reports (Harris et al., 2014;L.J. Carpenter and S. Reimann et al., 2014;Montzka and Reimann et al., 2011), but differs in two aspects. First, chemical loss is described by the new time independent fractional release factor $\overline{f}$ (Ostermöller et al., 2017) and, second, instead of the age spectrum $G$ with mean age $\Gamma$ for an inert tracer the normalized release weighted distribution function $G_N^{\#}$ with the corresponding mean release time $\Gamma^{\#}$ for a chemically active specie is used.

In order to apply this new formulation of EESC, the normalized release time distribution $G_N^{\#}$ and the time independent fractional release factors $\overline{f}$ (Ostermöller et al., 2017) for all relevant chlorine and bromine species need to be known. To our knowledge, so far only mean arrival times time $\Gamma^{*}$ for most species are available from the literature (Plumb et al., 1999). In the following we therefore show how the first moment of $G_N^{\#}$ (the mean release time $\Gamma^{\#}$) and $\overline{f}$ can be derived from the information available.

### 3.1. Deriving the mean release time.

The arrival time distribution $G^*$ (Plumb et al., 1999) and the release time distribution $G^\#$ used here are closely linked. By combining equation (4) and (5) it is easily shown that the sum of $G^*$ and $G^\#$ is the age spectrum G

$$G^\#(r, t') + G^*(r, t') = f(t') \cdot G(r, t') + \left(1 - f(t')\right) \cdot G(r, t') = G(r, t'). \tag{22}$$

In a similar way as for $G^\#$, a normalized arrival time distribution $G_N^*$ has been defined (Ostermöller et al., 2017)

$$G_N^*(r, t') = \frac{G^*(r, t')}{(1 - \overline{f}(r))}. \tag{23}$$

5     In a similar way as for the mean release time, a mean arrival time $\Gamma^*$ (Plumb et al., 1999) can be derived as the first moment of the arrival time distribution (4). In the arrival time distribution the age spectrum is not weighted with the release term $f(t')$ (as is the case for $G^\#$), but rather with the loss term $\left(1 - f(t')\right)$,

$$\int_0^\infty t' \cdot G_N^*(r, t') dt' = \Gamma^* \tag{24}$$

    $\Gamma^*$ has been calculated in a 2D model and is available for all relevant halocarbons based upon a parameterization (Plumb et al., 1999) as function of $\Gamma$ and lifetime $\tau$. $\Gamma^\#$, which is needed to calculate the release of inorganic halogen from a halocarbon

10     can be derived from the knowledge of $\Gamma$, $\Gamma^*$ and fractional release factors. To derive $\Gamma^\#$ we start with the relationship between $G^\#$ and $G^*$ (22).

Multiplying (22) with t' and integrating over all possible transit times yields

$$\int_0^\infty t' \cdot G(r, t') dt' = \int_0^\infty t' \cdot G^*(r, t') dt' + \int_0^\infty t' \cdot G^\#(r, t') dt'. \tag{25}$$

Replacing $G^*$ and $G^\#$ with their normalized distributions $G_N^*$ (23) and $G_N^\#$ (16) yields

$$\int_0^\infty t' \cdot G(r, t') dt' = \int_0^\infty t' \cdot \left(1 - \overline{f}(r)\right) \cdot G_N^*(r, t') dt' + \int_0^\infty t' \cdot \overline{f}(r) \cdot G_N^\#(r, t') dt'. \tag{26}$$

Extracting the transit time independent fractional release factors (Ostermöller et al., 2017) from the integrals yields

$$\int_0^\infty t' \cdot G(r, t') dt' = \left(1 - \overline{f}(r)\right) \cdot \int_0^\infty t' \cdot G_N^*(r, t') dt' + \overline{f}(r) \cdot \int_0^\infty t' \cdot G_N^\#(r, t') dt'. \tag{27}$$

15     All the integrals in (27) can be solved as they are the first moments of the respective distribution functions, thus mean age $\Gamma$ (9), mean arrival time $\Gamma^*$ (24) and mean release time $\Gamma^\#$ (14). (27) thus becomes

$$\Gamma = \left(1 - \overline{f}(r)\right) \cdot \Gamma^* + \overline{f}(r) \cdot \Gamma^\#. \tag{28}$$

Again we express $\bar{f}$ as a function of mean age $\Gamma$ instead of location r and then rearrange (28) to give an equation to calculate $\Gamma^\#$:

$$\Gamma^{\#} = \frac{\Gamma - (1 - \overline{f}(\Gamma)) \cdot \Gamma^{*}}{\overline{f}(\Gamma)}. \tag{29}$$

$\Gamma^{\#}$, which is the first moment of $G_N^{\#}$, can thus be derived based on the mean fractional release factor, mean age and $\Gamma^{*}$ for each compound. In the next section we now derive a formulation to calculate the mean fractional release factors $\overline{f}(\Gamma)$ from available fractional release factors.

## 3.2. Recalculating FRF values to yield time-independent mean FRF values $\overline{f}$.

For the calculation of mean release time $\Gamma^{\#}$, and also in the new formulation for EESC (21) the time independent mean fractional release factors $\overline{f}$ as derived by (Ostermöller et al., 2017) are needed.

$$\overline{f}(\Gamma) = \frac{\int_0^{\infty} \chi_0(t - t') \cdot G_N^{*}(\Gamma^{*}, t') dt' - \chi_{strat}(\Gamma, t)}{\int_0^{\infty} \chi_0(t - t') \cdot G_N^{*}(\Gamma^{*}, t') dt'} \tag{30}$$

The fractional release factors from the most recent WMO reports are largely based on observations from the time period 1996 to 2000 (Newman et al., 2007) and were derived using

$$f(\Gamma) = \frac{\int_0^{\infty} \chi_0(t - t') \cdot G(\Gamma, t') dt' - \chi_{strat}(\Gamma, t)}{\int_0^{\infty} \chi_0(t - t') \cdot G(\Gamma, t') dt'} \tag{31}.$$

In this formulation, the age spectrum for an inert tracer is used, which does not include chemical loss. Solving (31) for
$\chi_{strat}(\Gamma, t)$ and inserting this into (30) yields

$$\overline{f}(\Gamma) = \frac{\int_0^{\infty} \chi_0(t - t') \cdot G_N^{*}(\Gamma^{*}, t') dt' - (1 - f(\Gamma)) \cdot \int_0^{\infty} \chi_0(t - t') \cdot G(\Gamma, t') dt'}{\int_0^{\infty} \chi_0(t - t') \cdot G_N^{*}(\Gamma^{*}, t') dt'} \tag{32}$$

Equation (32) allows to convert fractional release factors $f$ calculated according to Newman et al. (2007) to time-independent values $\overline{f}$ according to Ostermöller et al. (2017). We derived $\overline{f}$ for every month of the period 1996 to 2000 from $f$ and then took the median of these values. The new $\overline{f}$ values and the spread of $\overline{f}$ values derived by the conversion during the different months of this period are shown in Tables 1 and 2. The spread is mostly very small, as temporal trends during this period were
small for many species. For the same reason, the $\overline{f}$ values derived in this way mostly do not differ very strongly from the $f$ values, as $f$ and $\overline{f}$ only differ due to tropospheric trends.

The fractional release factors used for the reference calculation presented in section 4 are those used by Velders and Daniel (2014), modified using equation (32) to be consistent with the new formulation given by Ostermöller et al. (2017). The mean release time $\Gamma^{*}$ has been calculated according to the parameterization of Plumb et al. (1999) also using their model lifetimes.
Plumb et al. (1999) used a 2D model for their study. Despite this, the stratospheric lifetimes derived from the model are in

overall good agreement to more recent model studies (Chipperfield et al., 2013). The sensitivity of the parameterization between mean release time and mean age to the stratospheric lifetimes is further discussed in section 4. For CFCs 114 and 115 which are not included in Plumb et al. (1999) we used stratospheric lifetimes from Ko et al. (2013), while the stratospheric lifetimes for HCFC-142b and the halons halon-1301, halon-1202 and halon-2402, which are also not included by Plumb et al.

(1999), are taken from Chipperfield et al. (2013). For those species included in Plumb et al. (1999) we used the species specific fit parameters, while for other species we used the averaged fit parameters reported in Plumb et al. (1999). The values for lifetimes used in the calculations and for both mean arrival time and mean release time as well as the mean fractional release factors $\bar{f}$ are given for all species used in this calculation in Tables 1 and 2.

**4. Temporal evolution of EESC and implications for recovery to 1980 benchmark values.**

The new formulation of EESC (21) uses a loss weighted transit time distribution, the release time distribution, and different fractional release factors from those used in the formulation (Newman et al., 2007). The new fractional release factors are based on the formulation suggested by Ostermöller et al. (2017) and have been derived from available fractional release factors (see section 3.2),. No method to calculate the release time distribution is available so far. Both the mean release time $\Gamma^{\#}$ (first

moment of the distribution) and the shape of the release time distribution $G_N^{\#}$ need to be known in order to use this distribution for the calculation of the propagation of tropospheric trends into the stratosphere. The age spectrum for an inert tracer, $G$, is commonly described by an inverse Gaussian function with a parameterization of the width as function of mean age (Hall and Plumb, 1994;Schauffler et al., 2003;Newman et al., 2007). As no such parameterization has yet been established for $G_N^{\#}$ we have assumed that the general shape of $G_N^{\#}$ is similar to that of $G$, with $\Gamma^{\#}$ instead of $\Gamma$ as mean value. The sensitivity of our

calculations to these assumptions is discussed in section 4.2.

**4.1. Comparison of different EESC formulations**

As already mentioned, new time-independent fractional release factors and the release time distribution are needed for our new formulation of EESC. The release time distribution is approximated assuming the form of an inverse Gaussian with a species specific first moment $\Gamma^{\#}$ and a width of $\lambda = \frac{\Delta^{\#^2}}{\Gamma^{\#}} = 0.7 \ years$.

The new time-independent fractional release factors are based on the concept of arrival time distribution (Plumb et al., 1999). Ostermöller et al. (2017) showed that using this concept, fractional release factors can be calculated, which are independent of time, as long as stratospheric transport or photochemistry remain unchanged. More specifically, these fractional release factors are independent of the tropospheric trend of the respective species. We have recalculated fractional release factors used in the most recent ozone assessment report (Harris et al., 2014) to be consistent with the new formulation of fractional release. The

fractional release factors commonly used are largely based on observations (Newman et al., 2007), except for the

hydrochlorofluorocarbons HCFC-141b and HCFC-142b (Daniel et al., 1995) (see Tables 1 and 2). Other observation based fractional release factors have been presented by Laube et al. (2013). The uncertainty due to the use of different fractional release factors, different emissions and different lifetimes have been discussed in details by Velders and Daniel (2014) . Here, we focus on the uncertainties due to the suggested new formulation for the calculation of EESC. Using these new FRF values

and the mean arrival time $\Gamma^*$ based on the available parameterization (Plumb et al., 1999), we have calculated values for $\Gamma^{\#}$ for all relevant chlorine and bromine containing source gases (see Tables 1 and 2).

Figures 3 and 4 show the calculation according to (21) using the new time independent FRF values for 3 and 5.5 years of mean age, respectively, and compare it with the calculation applying formulation (3) using the FRF values of the ozone assessment reports (Harris et al., 2014). All values given here are mole fractions given in ppt, which is equivalent to pmol/mol. The values

are also summarized in Tables A1 and A2. The tropospheric time series and the future projection used for this calculation are based on Velders and Daniel (2014), where updated lifetimes (Ko et al., 2013) and assumptions on future emissions have been used as basis for the projection of tropospheric time series. For a mean age of air of 3 years, as used for mid-latitudes, there are significant differences between the two methods used for calculation of EESC (see Figure 3). In the case of our new formulation, there is a longer time lag between the troposphere and the arrival of the inorganic halogen in the stratosphere.

The tropospheric halogen loading was increasing strongly during the time before 1980, and therefore EESC at that time was dominated by air masses that had a lower halogen content. As a consequence, we calculate 1980 EESC levels in the mid latitude lower stratosphere which are about 90 ppt lower (see Table 3 for details) than using the EESC formulation according to Newman et al. (2007). During the recovery phase of stratospheric halogen loading, temporal trends of halogenated source gases in the troposphere will be negative; EESC will thus be dominated by air masses with higher chlorine content and is

higher in our new formulation. In combination with the lower level of EESC, which must be attained for recovery, a significantly later recovery date is calculated. According to our calculation, mid latitude lower stratospheric EESC levels will return to 1980 values in 2060 only, which is more than 10 years later than the recovery date of 2049 calculated using the current method (Velders and Daniel, 2014).

For polar winter conditions (5.5 years of mean age) shown in Figure 4, the recovery date calculated here is 2077, relative to a

value of 2076 derived based on the currently used method using the same scenario (Velders and Daniel, 2014). The reason that only a very minor change is calculated for polar winter conditions is that under these conditions nearly all source gases are converted to their inorganic form and the differences between the age spectrum and the release time distribution become very small.

## 4.2. Sensitivity discussion and tests

As mentioned above, we will concentrate on the sensitivity of the new EESC method on the limited knowledge of the new release time distribution $G_N^{\#}$ and on the new fractional release factors (Ostermöller et al., 2017) used here. We have therefore

performed sensitivity calculations to evaluate the sensitivity of our results on the changed fractional release factors and on the uncertainty in the knowledge of the release time distribution $G_N^{\#}$.

**Sensitivity to new fractional release factors**

To evaluate the changes due to the changes in fractional release factors, we use our new release time distribution $G_N^{\#}$, but use the same fractional release factors as in previous studies (Velders and Daniel, 2014;Harris et al., 2014). The comparison for a mean age of 3 years is shown in Figure 5 and Table 3 (bottom row). The estimated recovery year is 2058, instead of 2060 using our new fractional release factors. The change in maximum EESC is also small with a value of 1909 using our new fractional release factors and 1895 using the fractional release factors as in Velders and Daniel (2014). For 5.5 years of mean age (not shown), the same recovery date is calculated (2077) with about 20 ppt lower EESC during the maximum using the unmodified fractional release factors from Velders and Daniel (2014). The new time-independent fractional release factors are derived from the fractional release factors presented by Newman et al. (2007) as described in the Appendix. For this, a correction needs to be applied (see Appendix) based on the year of the measurements from which the fractional release factors have been derived. The fractional release factors used by Newman et al. (2007) were derived from the measurements taken during the STRAT campaign (1996), the POLARIS campaign (1997) and from the SOLVE (Schauffler et al., 2003) campaign (1999-2000). We converted the fractional release factors assuming that they were taken during the time period 1996-2000, with the exception of HCFC-s 141b and 142b, where we used the values given in Velders and Daniel (2014). We performed the conversion for every month of this period. The median of all values was taken as the best estimate for the new time-independent fractional release factors. We also derived a variability which was below 1% for most species. This variability is also presented in Tables 1 and 2 together with the new fractional release factors. As a sensitivity test, we performed the EESC calculation by shifting all fractional release values up or downwards by 1σ. Varying the new fractional release factors whitin this uncertainty range resulted in an upward (increased fractional release) or downward (decreased fractional release) shift of EESC by about 14 ppt during the maximum of EESC for the 3 year mean age calculation. The changes in the calculated recovery years were less than 0.2 years. For 5.5 years of mean age, the variation of the fractional release factors lead to even smaller changes. Overall, we conclude that the changes in the fractional release factors are rather small in comparison to the overall changes. For polar winter conditions, the calculated changes were in all cases very small. The significant differences in recovery dates for the mid latitude lower stratosphere presented above are thus mainly due to the new release time distribution $G_N^{\#}$.

**Sensitivity to the shape of the new release time distribution $G_N^{\#}$**

The new release time distribution $G_N^{\#}$ has not been calculated from models to our knowledge. We therefore have to make assumptions on the shape and the width of $G_N^{\#}$. In the calculation presented above, we have assumed that the shape is similar

as for $G$, i.e. an inverse Gaussian function. We then made the assumption that the width can be described in a similar way as a function of the first moment, i.e. using a constant factor $\lambda = \frac{\Delta^2}{\Gamma} = \frac{\Delta^{\#2}}{\Gamma^{\#}} = 0.7 \ years$ (Hall and Plumb, 1994; Engel et al., 2002). In order to test the sensitivity of our results to these assumptions, we have varied this parameter between values of 0 and 2 years in the calculation of $G_N^{\#}$, while retaining the first moment, i.e. $\Gamma^{\#}$. The extreme case of 0 would mean that the release

time distribution $G_N^{\#}$ collapses to one single transit time or lag time, i.e. $\Gamma^{\#}$ and that no mixing occurs during the transport in the stratosphere. In this case, stratospheric inorganic chlorine is simply derived by time shifting the tropospheric time series of a source gas by the time lag $\Gamma^{\#}$ and multiplying it with the fractional release factor.

The scenarios with $\lambda$ equal to 0 years (no mixing case), 0.7 years (reference case) and 2 years (strong mixing case) are compared in Figure 6 and 7 for mean age values of 3 and 5.5 years. It is obvious, that the calculation of the recovery year is

not very sensitive to the width of the release time distribution $G_N^{\#}$. In the case of the pure lag time calculation assuming no mixing ($\lambda = 0 \ years$) recovery is about 1 year later for both 3 and 5.5 years of mean age compared to the reference case ($\lambda = 0.7 \ years$). In the case of a very wide spectrum assuming strong mixing ($\lambda = 2 \ years$), recovery is expected about 1.5 years earlier at the mean age level of 3 years and 2.5 years earlier at the mean age level of 5.5 years.

The overall range of the calculated recovery dates is 2.2 years in the case of 3 years mean age and 3.8 years in the case of 5.5

years of mean age. The recovery dates and the maximum values of EESC calculated under the different assumptions are compared in Table 4. This rather small dependence on the width of the applied transit time distribution even for the assumption of extreme cases is due to two factors. Firstly, the deviation of tropospheric trends from linearity in the years prior to the reference year of 1980 and during the recovery phase after 2030 are rather small, in which case the propagation values becomes independent of the shape of the distribution (Hall and Plumb, 1994) and only depend on the first moment, i.e. the mean release

time $\Gamma^{\#}$. Secondly, during the period of the maximum EESC, tropospheric trends are rather small overall and thus the trend correction becomes rather small. Therefore maxiumum differences are below 50 ppt both for 1980 values and for the maximum EESC values for 3 years of mean age and 73 ppt for 5.5 years of mean age. The largest dependence on the parameterization is thus derived for 5.5. years of mean age during the maximum of EESC, as tropospheric data from a large time period need to be taken into account here and there is significant non-linearity in the trends. In all cases, the lag time calculation yields higher

EESC during the maximum, as would be expected.

**Sensitivity to the mean release time derived from mean age and stratospheric lifetime.**

Another source of uncertainty is, that the stratospheric lifetime of the individual compounds needs to be known in order to calculate the mean arrival time $\Gamma^*$ from which the mean release time $\Gamma^{\#}$ is derived (29). We tested the sensitivity of our calculation to that by systematically increasing all lifetimes by 20% or decreasing them by 20% (see Table 5) in the

parameterization given by (Plumb et al., 1999). This results in different mean arrival time $\Gamma^*$ and mean release time $\Gamma^{\#}$. Even if such rather large changes would go in the same direction for all species, the impact on our calculations is rather small. For

years of mean age, the calculated maximum in EESC varied by only 6 ppt and the calculated recovery date varied by 2.5 years. For 5.5 years of mean age the effect is even smaller with variation in maximum EESC of less than 1 ppt and a variation of less than half a year in the recovery date. This very small sensitivity at 5.5 years of mean age is due to most fractional release factors being close to 1 under these conditions. The reason for this small effect is that next to the parameterization of the mean

arrival time, the fractional release factor determines the mean release time. Therefore, the sensitivity of the mean arrival time to the assumed lifetime in our calculation is quite low. Consequently, the influence on the derived EESC is also rather small. Despite this rather low sensitivity, it should be noted that the parameterization is derived from a 2D model. The relationship between mean age, age spectrum and chemical loss should be explored in state-of-the-art 3 D models, which have a better representation of stratospheric transport processes.

**4.3. Comparison of EESC formulations with model calculations of inorganic halogen loading.**

In order to evaluate our new formulation of EESC we have compared the results of our calculations with the inorganic halogen loading calculated from two comprehensive three-dimensional atmospheric chemistry models. Due to expected long-term changes in mean age on a given pressure level associated with the simulated changes in the Brewer-Dobson circulation (e.g. Butchart, 2014;Austin and Li, 2006), changes in fractional release factors on mean age levels are also observed in free running

model calculations (Douglass et al., 2008;Li et al., 2012b). We have therefore compared our new formulation of EESC to model calculations with changing and with annually repeating ('fixed') dynamics. To compare the new formulation with the formulation by (Newman et al., 2007), we used a model simulation from the TOMCAT model (Chipperfield et al., 2017), which was driven by a repeated meteorology, in this case for the year 1980. Effects due to changing dynamics, which are not included in the concept of EESC, will thus not impact this calculation, making it an ideal test bed for comparison of the two

formulations. For long term changes, we have used model results from the EMAC model (Jöckel et al., 2016), which includes expected changes in stratospheric transport. As in general the relationship between mean age and $Cl_y$ is very different for different models (Waugh et al., 2007), a direct comparison between EESC and ESC (Equivalent stratospheric Chlorine, calculated from model $Cl_y$ and $Br_y$ using the same sensitivity parameters for bromine as with EESC) is not meaningful, as differences may be due to different fractional release factors between models and observations. Instead we have used fractional

release values derived from the models for 3 years of mean age and used these for the calculation of EESC using the formulations by Newman et al. (2007) and from this work. Fractional release factors were calculated from the model data using the methods of Newman et al. (2007) and Ostermöller et al. (2017) for this work. The fractional release factors were calculated for the year 2000, in order to be consistent with the observation based fractional release factors, which were derived mainly for the year 2000. To the EESC calculated in this way we added simulated inorganic chlorine and bromine at the

tropical tropopause, which we propagated as an inert tracer. VSLS (very short lived substances) were treated in a similar way, using their tropical tropopause values as input, as loss in the troposphere cannot be neglected for these species. As no global stratospheric lifetimes are available for these species, it is not possible to apply the new formulation. Therefore, the VSLS

were treated using the method of Newman et al. (2007). The differences are negligible, as the VSLS have rather slow long-term trends and both methods yield nearly identical results, as also discussed in Ostermöller et al. (2017). As some loss of $CH_3Cl$ and $CH_3Br$ occurs during the transport in the troposphere to the tropical tropopause, we have also used the time series of these two gases at the tropical tropopause rather than at the surface in these calculations.

## 5  Comparison to fixed dynamics model calculations

For comparison of the two formulations to model calculations with fixed dynamics, we used a TOMCAT model run (Chipperfield et al., 2017), which was driven by repeated 1980 meteorology. Output from this model run is available from the years 1960 - 2016. The fractional release factors derived from the model for the northern hemisphere are significantly higher as a function of mean age than the observed fractional release values. Southern hemispheric fractional release values for 3 years of mean age showed better agreement with observation derived fractional release factors (Newman et al., 2007). For this reason we compared simulated ESC from the southern hemisphere with EESC calculated using our new formulation and the formulation by (Newman et al., 2007), in both cases using fractional release values derived for the year 2000 model results. Figure 8 shows the comparison between the modelled ESC and EESC for a mean age of 3 years calculated as described above, including all bromine and chlorine species included in the model and also including inorganic chlorine and bromine entering the stratosphere. As the differences between the two formulations of EESC are most pronounced for 3 years of mean age, we show this comparison for 3 years of mean age only. A much better agreement is observed when applying the new formulation than using the formulation by Newman et al. (2007), due to the improved treatment of the combined influence of transport and mixing on chemical loss. Remaining discrepancies between model ESC and EESC are most probably due to an imperfect parameterization of the loss time distribution $G^{\#}$.

## 20  Comparison to model calculations with varying dynamics

Under changing stratospheric dynamics (e.g.Butchart, 2014), it is expected that fractional release factors at a given mean age level will change (Douglass et al., 2008;Li et al., 2012b;Ostermöller et al., 2017). Therefore, the inorganic halogen loading as a function of mean age would be expected to change even if all source gases remained constant in time. Under such conditions, EESC is not expected to follow ESC on a given mean age level. To estimate the validity of EESC as a proxy for inorganic halogen loading of the stratosphere, we have compared our new formulation to a free running chemistry climate model simulation. We used data from the EMAC model simulation RC2-base-04 from the ESCiMo project for this (Jöckel et al., 2016). This simulation covers the 1950–2100 time frame with simulated sea surface temperatures and sea ice contents. As described above, we again calculated fractional release factors from the model in order to have results which are internally consistent. Northern Hemisphere fractional release factors for the year 2000 are in good agreement with observation based fractional release factors(Newman et al., 2007;Laube et al., 2013) and therefore we used northern hemispheric data for this

comparison. In addition to comparing ESC on a fixed mean age level we also compared ESC on a fixed pressure level to our new formulation of EESC. A similar comparison has been presented in Shepherd et al. (2014), who compared model ESC on a fixed pressure level to EESC on a fixed mean age level (using the formulation of Newman et al. (2007)), showing good agreement. Figure 9 compares the time evolution of EESC for 3 years of mean age with model ESC at the 60 hPa level

(corresponding to 3 years of mean age in the year 2000) and model ESC at 3 years of mean age. The year 2000 and the corresponding level of 60 hPa were chosen, as we also evaluated fractional release factors in the year 2000 of the model run. As expected, ESC at 3 years of mean age deviates systematically from EESC, especially in the future when fractional release evaluated on a mean age surface changes significantly in the model. The agreement with ESC on a fixed pressure level is however much better. In this comparison EESC at 3 years of mean age would slightly overestimate ESC on a pressure level in

the future and significantly underestimate ESC on a mean age level. The exact magnitude of changes in stratospheric dynamics is highly uncertain but it has been shown that ESC evaluated at pressure levels is a good proxy to describe the influence of halogens on the ozone column (Shepherd et al., 2014;Eyring et al., 2010). Based on the much better agreement of EESC with ESC at pressure levels, we conclude that EESC is a reasonable proxy for the effect of halogen loading on stratospheric ozone, given the overall high uncertainties associated to the future evolution of stratospheric dynamics.

**5. Conclusions and outlook**

We have shown that for the calculation of the propagation of chlorine and bromine source gases with photochemical loss, different transit time distributions must be used to calculate the amount of organic or inorganic chlorine present at a given mean age level. First, treating the propagation of these tracers with the age spectrum for an inert tracer leads to fractional release values, which show a strong temporal variability in case of large tropospheric trends of the respective gas (Ostermöller

et al., 2017). Fractional release factors, which are independent of the tropospheric trends (Ostermöller et al., 2017), must be used to correctly describe the fraction that has been transferred from the organic source gas to the inorganic form and can then influence ozone chemistry. Secondly, changes in tropospheric mixing ratios lead to changes in stratospheric inorganic halogen with a time delay that is longer than the mean age, which describes the propagation of an inert tracer. This can be described by a modified transit time distribution, in which the transit times from the classical age spectrum are weighted with the chemical

loss during this transport time. We suggest the term "release time distribution" for this modified tracer specific transit time distribution.

We developed a new formulation of EESC, which uses the release time distribution and time-independent fractional release factors calculated by the method of Ostermöller et al. (2017). This approach more accurately represents the amount of $Cl_y$ and $Br_y$ in the stratosphere from tropospheric source gas concentrations and fractional release factors, as shown in comparison to

a model calculation with annually repeating dynamics. We have shown that the long-term evolution of equivalent stratospheric chlorine (ESC, i.e. inorganic chlorine and bromine, the latter weighted in a similar way as in EESC to reflect the higher efficiency of bromine to ozone depletion) in the model deviates substantially from our calculation of EESC in a long-term model calculation with varying dynamics. However, we have also shown that the new formulation of EESC is a reasonable

proxy for the evolution of inorganic halogen loading on a given pressure level. We therefore conclude that EESC is a reasonable proxy for future halogen impact on ozone. We suggest our new method should be adopted to calculate EESC and to estimate the time of recovery of inorganic halogen to 1980 values under otherwise unchanged conditions. This will lead to a delay of about 10 years in calculated EESC recovery in the lower mid-latitude stratosphere (mean age of 3 years) compared to the

formulation currently used (Velders and Daniel, 2014;Newman et al., 2007), and also applied in the WMO ozone assessment reports (Harris et al., 2014;L.J. Carpenter and S. Reimann et al., 2014). If all other factors were unchanged, in particular stratospheric dynamics, the recovery of the mid-latitude lower stratosphere to unperturbed values of chlorine and bromine would thus also expected to take about 10 years longer than previously estimated using EESC based on the formulation by Newman et al. (2007). As current climate models consistently predict an acceleration in the Brewer-Dobson circulation

(Butchart, 2014), this will have an impact on the temporal evolution of inorganic halogen loading of the stratosphere. These expected changes in the Brewer-Dobson circulation would result in an earlier recovery of ozone at mid- and high latitudes (Eyring et al., 2010). These changes are not included in the concept of EESC. However, we have shown that EESC is a reasonable proxy for ESC when ESC is evaluated at constant pressure level. In addition to this, increases in the concentrations of $N_2O$ and short-lived chlorine-containing halocarbons may further influence the recovery of the ozone layer, possibly leading

to a later recovery (Hossaini et al., 2015b;Hossaini et al., 2015a;Chipperfield, 2009). The changes due to application of our new method for 5.5 years of mean age (representative of polar winter conditions) are rather small compared to the formulation suggested by Newman et al. (2007), as nearly all halogen is released under these conditions and the difference between age spectrum and release time distribution becomes small.

The two changes relative to the currently used formulation for EESC are the use of new time-independent fractional release

factors and of the release time distribution. We have shown that the new time-independent fractional release factors do not differ very much from the fractional release factors currently used (Harris et al., 2014;Velders and Daniel, 2014), as they were derived during a period of rather small tropospheric trends for many species. Consequently, the projected EESC recovery dates vary by 2 years or less depending on which fractional release factors are used. We have also shown that the calculation of the recovery date shows some sensitivity to the assumed width of the release time distribution, with variations of about 2 years for

the mid-latitude calculations and 3.5 years for the high latitude case. Varying the stratospheric lifetimes assumed for the calculation of the loss weighted transit distribution $G^{\#}$ a similar influence on the projected recovery dates for 3 years mean age (mid-latitude conditions) and virtually no effect for 5.5 years (high-latitude conditions) is derived. In general, the maximum EESC level is more sensitive to variations in the assumed width of the release time distribution than to the stratospheric lifetimes assumed in the calculation of the mean release time. The strongest dependence on the assumed width is observed

during the maximum of EESC levels, especially for polar winter conditions, as tropospheric trends were strongly non-linear during that time. A more realistic description of the shape of the release time distribution would improve especially the prediction of EESC during its maximum. Age spectra for inert tracers in models for atmosphere and ocean have been derived from pulse experiments using tracers without chemical loss (Haine et al., 2008;Li et al., 2012c;Li et al., 2012a;Ploeger and Birner, 2016). For the derivation of the release time distribution and improved information on mean release time such pulse

experiments for tracers with chemical loss should be performed. Such calculations are only available based on a rather old 2D model (Plumb et al., 1999) and should be repeated with state-of-the-art models. These release time distributions will be specific for each tracer, but should generally be similar for species with similar lifetimes and similar loss processes. As we have shown good agreement with model calculations using the parameterization suggested here, we do not expect a re-evaluation of the loss weighted age spectra to lead to large changes. As the suggested reformulation of EESC does not affect the principal behavior of the temporal evolution of EESC, we do not expect this reformulation to lead to substantial changes, which could impact the changes of studies using EESC except for those which have used EESC to project EESC recovery. For studies using EESC as a proxy for the halogen loading, e.g. in comparison to ozone time series, the new formulation of EESC suggested here should nevertheless be used, as the timing of the recovery especially for mid latitudes is significantly different than in previous estimates.

*Data availability:* The work is based on the scenario developed by (Velders and Daniel, 2014). We have used the scenario "All_parameters_SPARC2013_mostlikely_mc.dat" given in the Appendix to that paper and available as supplement. For data availability of the EMAC simulation results we refer to Jöckel et al., (2016). The TOMCAT model results are available by emailing Martyn Chipperfield.

*Author contribution:.* Andreas Engel has performed most of the calculations in the manuscript, has written the manuscript and has developed the ideas presented in the manuscript together with the Jennifer Ostermöller and Harald Bönisch in the frame of many open discussions. Jennifer Ostermöller and Harald Bönisch have both participated in the discussion and preparation of the manuscript. Jennifer Ostermöller has performed the calculations for the comparison with model data (section 4.3.). Sandip Dhomse and Martyn Chipperfield have provided TOMCAT model results. Patrick Jöckel is PI of the ESCiMo project, conducted the EMAC model simulations and provided the corresponding data. All authors were involved in the final revision of the manuscript.

*Acknowledgements.* This work was partly supported by DFG Research Unit 1095 (SHARP) under project numbers EM367/9-1 and EN367/9-2. The EMAC simulations have been performed at the German Climate Computing Centre (DKRZ) through support from the Bundesministerium für Bildung und Forschung (BMBF). DKRZ and its scientific steering committee are gratefully acknowledged for providing the HPC and data archiving resources for the consortial project ESCiMo (Earth System Chemistry integrated Modelling). The TOMCAT simulations were performed on the Archer and Leeds ARC supercomputers. We thank Wuhu Feng (NCAS) for help with the model. We thank Tanja Schuck and Joachim Curtius for comments on the manuscript.

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

**Table 1:** Mean arrival time $\Gamma^*$, mean release time $\Gamma^\#$, $f$ and $\bar{f}$ for all relevant long lived chlorine and bromine species for a mean age of 3 years. The time independent fractional release factors were derived using equation (32). The measurements from which the original fractional release factors were derived are from the period 1996-2000. We calculated the conversion using eq. (32) for every month of this period for all species, except for HCFC-141b and 142b (see text). The 1σ variability of the converted fractional release factors is also shown.

| | Stratospheric lifetime used for $\Gamma^*$ [years] | mean arrival time $\Gamma^*$ [years] | mean release time $\Gamma^\#$ [years] | fractional release factor f | time independent fractional release factor $\bar{f}$ | 1 σ variability of recalculated $\bar{f}$ in % |
|---|---|---|---|---|---|---|
| | | Mean age $\Gamma$ of 3 years | | | | |
| CFC-11 | 55.7[a] | 1.5 | 4.7 | 0.47[d] | 0.47 | 0.23 |
| CFC-12 | 102[a] | 2.0 | 6.2 | 0.23[d] | 0.24 | 0.22 |
| CFC-113 | 84.5[a] | 1.9 | 5.7 | 0.29[d] | 0.30 | 0.69 |
| CFC-114 | 189[b] | 2.2 | 8.7 | 0.12[d] | 0.13 | 0.11 |
| CFC-115 | 1020[b] | 2.5 | 12.1 | 0.04[d] | 0.07 | 0.45 |
| CCl$_4$ | 48.4[a] | 1.4 | 4.3 | 0.56[d] | 0.56 | 0.08 |
| CH$_3$CCl$_3$ | 48.8[a] | 1.4 | 4.1 | 0.67[d] | 0.61 | 1.93 |
| HCFC-22 | 217[a] | 2.6 | 5.6 | 0.13[d] | 0.15 | 0.09 |
| HCFC-141b | 73[a] | 1.8 | 5.4 | 0.34[e] | 0.34 | n.a.[f] |
| HCFC-142b | 212[c] | 2.2 | 6.8 | 0.17[e] | 0.17 | n.a.[f] |
| halon-1211 | 39.5[a] | 1.3 | 4.0 | 0.62[d] | 0.65 | 0.4 |
| halon-1202 | 15.3[c] | 0.5 | 5.0 | 0.62[d] | 0.67 | 2.79 |
| halon-1301 | 70.8[c] | 1.7 | 6.0 | 0.28[d] | 0.32 | 1.34 |
| halon-2402 | 33.8[c] | 1.2 | 4.0 | 0.65[d] | 0.66 | 0.55 |
| CH$_3$Br | 40.2[a] | 1.3 | 4.1 | 0.60[d] | 0.60 | 0.11 |
| CH$_3$Cl | 63.7[a] | 1.9 | 4.4 | 0.44[d] | 0.44 | 0[g] |

[a] stratospheric lifetime from the 2D model used by Plumb et al. (1999)
[b] total atmospheric lifetime taken from recommendations in SPARC lifetime assessment (Ko et al., 2013), as tropospheric loss is negligible.
[c] stratospheric lifetime taken from modelling work for SPARC lifetime assessment (Chipperfield et al., 2013).
[d] fractional release values based on Newman et al. (2007)
[e] fractional release value based on parameterization given in footnote to table 2 in Velders and Daniel (2014).
[f] not applicable, as the fractional release values have not been recalculated.
[g] no variability was derived as there is no trend in the tropospheric reference data applied.

**Table 2:** Mean arrival time $\Gamma^*$, mean release time $\Gamma^\#$, $f$ and $\bar{f}$ for all relevant long lived chlorine and bromine species for a mean age of 5.5 years. In the case of fractional release factor of 1, there is no remaining organic fraction and the concept of mean arrival time $\Gamma^*$is not applicable (n.a.). Inorganic chlorine can then be treated in a similar way as an inert tracer, using mean age $\Gamma$. The time independent fractional release factors were derived using equation (32). The measurements from which the original fractional release factors were derived are from the period 1996-2000. We calculated the conversion using eq. (32) for every month of this period for all species, except for HCFC-141b and 142b (see text). The 1σ variability of the converted fractional release factors is also shown.

| | Stratospheric lifetime used for $\Gamma^*$[years] | mean arrival time $\Gamma^*$ [years] | mean release time $\Gamma^\#$ [years] | fractional release factor f | time independent fractional release factor $\bar{f}$ | 1 σ variability of recalculated $\bar{f}$ in % |
|---|---|---|---|---|---|---|
| | | Mean age $\Gamma$ of 5.5 years | | | | |
| CFC-11 | 55.7[a] | 1.8 | 5.5 | 0.99[d] | 0.99 | 0.01 |
| CFC-12 | 102 [a] | 3.0 | 5.9 | 0.86[d] | 0.87 | 0.11 |
| CFC-113 | 84.5 [a] | 2.7 | 5.8 | 0.90[d] | 0.91 | 0.26 |
| CFC-114 | 189[b] | 3.6 | 8.3 | 0.40[d] | 0.41 | 0.28 |
| CFC-115 | 1020[b] | 4.4 | 10.1 | 0.15[d] | 0.20 | 0.58 |
| CCl₄ | 48.4 [a] | n.a. | 5.5 | 1.00[d] | 1.00 | 0.00 |
| CH₃CCl₃ | 48.8 [a] | 1.7 | 5.6 | 0.99[d] | 0.99 | 0.13 |
| HCFC-22 | 217 [a] | 4.3 | 7.0 | 0.41[d] | 0.44 | 0.16 |
| HCFC-141b | 73 [a] | 2.5 | 5.8 | 0.90[e] | 0.90 | n.a.[f] |
| HCFC-142b | 212[c] | 3.7 | 6.5 | 0.65[e] | 0.65 | n.a.[f] |
| halon-1211 | 39.5 [a] | n.a. | 5.5 | 1.00[d] | 1.00 | 0.00 |
| halon-1202 | 15.3[c] | n.a. | 5.5 | 1.00[d] | 1.00 | 0.00 |
| halon-1301 | 70.8[c] | 2.2 | 6.2 | 0.80[d] | 0.83 | 0.77 |
| halon-2402 | 33.8[c] | n.a. | 5.5 | 1.00[d] | 1.00 | 0.00 |
| CH₃Br | 40.2 [a] | 1.5 | 5.5 | 0.99[d] | 0.99 | 0.01 |
| CH₃Cl | 63.7 [a] | 2.9 | 5.8 | 0.91[d] | 0.91 | 0.00[g] |

[a] stratospheric lifetime from the 2D model used by Plumb et al. (1999)
[b] total atmospheric lifetime taken from recommendations in SPARC lifetime assessment (Ko et al., 2013), as tropospheric loss is negligible.
[c] stratospheric lifetime taken from modelling work for SPARC lifetime assessment (Chipperfield et al., 2013).
[d] fractional release values based on Newman et al. (2007)
[e] fractional release value based on parameterization given in footnote to table 2 in Velders and Daniel (2014).
[f] not applicable, as the fractional release values have not been recalculated.
[g] no variability was derived as there is no trend in the tropospheric reference data applied.

*Table 3:* Recovery years for EESC to return to 1980 values and maximum EESC values using our new formulation and the current formulation (Newman et al., 2007), as shown in Figures 3 and 4. In all cases the width Δ is parametrized based a values of $\lambda = \frac{\Delta^2}{\Gamma} = \frac{\Delta^{\#2}}{\Gamma^{\#}} = \mathbf{0.7\ years}$ (see text for an explanation of the parameterization). In the new formulation, the time-independent fractional release values and the values for $\mathbf{\Gamma^{\#}}$ **shown in Tables A1 and A2** have been used. The results in the bottom row were derived using the same fractional release factors as in (Velders and Daniel, 2014), instead of the new time-independent fractional release factors, as shown in Figure 5 for 3 years of mean age. Decimal places are not meant to imply that recovery dates can be calculated to this accuracy rate, but are only given in order to show the sensitivity of the calculations to different parameters. All values given here are mole fractions given in ppt, which is equivalent to pmol/mol.

| | 3 years mean age | | | 5.5 years mean age | | |
|---|---|---|---|---|---|---|
| | EESC 1980 [ppt] | EESC max [ppt] | 1980 recovery date | EESC 1980 [ppt] | EESC max [ppt] | 1980 recovery date |
| New formulation, time independent fractional release factors (Table A1 and A2) and G#. | 1065 | 1909 | 2059.9 | 2070 | 4107 | 2077.5 |
| Current formulation, fractional release factors as in Velders and Daniel (2014), age spectrum G. | 1154 | 1932 | 2048.6 | 2085 | 4102 | 2075.7 |
| New formulation using G#, but unchanged fractional release factors as in Velders and Daniel (2014) | 1070 | 1895 | 2057.8 | 2066 | 4088 | 2077.1 |

*Table 4:* Recovery years for EESC to return to 1980 values and maximum EESC values using different assumptions on the width of the release time distribution G# as shown in Figure 6 and 7. In all cases the general shape was assumed to be an inverse Gaussian function with different parameterizations of the width Δ, based on different values of $\lambda = \frac{\Delta^{\#2}}{\Gamma^{\#}}$ (see text for an explanation of the parameterization). The case of the pure lag time calculation is equal to $\lambda = \mathbf{0}$ years , i.e. the effect of mixing is completely ignored, $\lambda = \mathbf{2\ years}$ represents a case with strong mixing and a broad transit time distribution, while , $\lambda = \mathbf{0.7\ years}$ corresponds to our reference calculation. In all calculations, the same time-independent fractional release values have been used, as shown in Tables A1 and A2. Decimal places are not meant to imply that recovery dates can be calculated to this accuracy rate, but are only given in order to show the sensitivity of the calculations to different parameters. All values given here are mole fractions given in ppt, which is equivalent to pmol/mol.

| | 3 years mean age | | | 5.5 years mean age | | |
|---|---|---|---|---|---|---|
| | EESC 1980 [ppt] | EESC max [ppt] | 1980 recovery date | EESC 1980 [ppt] | EESC max [ppt] | 1980 recovery date |
| $\lambda = 0.7$ years reference | 1065 | 1909 | 2059.9 | 2070 | 4107 | 2077.5 |
| $\lambda = 2$ years , strong mixing | 1101 | 1913 | 2058.5 | 2149 | 4103 | 2075.0 |

| $\lambda = 0$, no mixing | 1055 | 1941 | 2060.7 | 2046 | 4176 | 2078.8 |
|---|---|---|---|---|---|---|

*Table 5:* Recovery years for EESC to return to 1980 values and maximum EESC values varying the stratospheric lifetimes in the calculations of the mean arrival time $\mathbf{\Gamma}^*$ from which the mean release time time $\mathbf{\Gamma}^{\#}$ is calculated according to (A8). In these calculations, the width of the release time distribution $G^{\#}$ was kept constant at $\lambda = \frac{\Delta^{\#2}}{\Gamma^{\#}} = \mathbf{0.7}$ years. The lifetimes have been varied systematically up- and downward by 20% for this sensitivity test (see text for explanations). The reference case is the same as shown in Table 1. For the high $\tau$ and low $\tau$ calcuations the stratospheric lifetimes of all species have been increased systematically by 20% upwards and downwards respectively, when calculating the mean arrival times and the mean release times. Decimal places are not meant to imply that recovery dates can be calculated to this accuracy rate, but are only given in order to show the sensitivity of the calculations to different parameters. All values given here are mole fractions given in ppt, which is equivalent to pmol/mol.

|  | 3 years mean age | | | 5.5 years mean age | | |
|---|---|---|---|---|---|---|
|  | EESC 1980 [ppt] | EESC max [ppt] | 1980 recovery date | EESC 1980 [ppt] | EESC max [ppt] | 1980 recovery date |
| Reference | 1064 | 1909 | 2059.9 | 2070 | 4107 | 2077.5 |
| High $\tau$ | 1073 | 1912 | 2058.8 | 2071 | 4107 | 2077.3 |
| low $\tau$ | 1054 | 1906 | 2061.3 | 2068 | 4107 | 2077.6 |

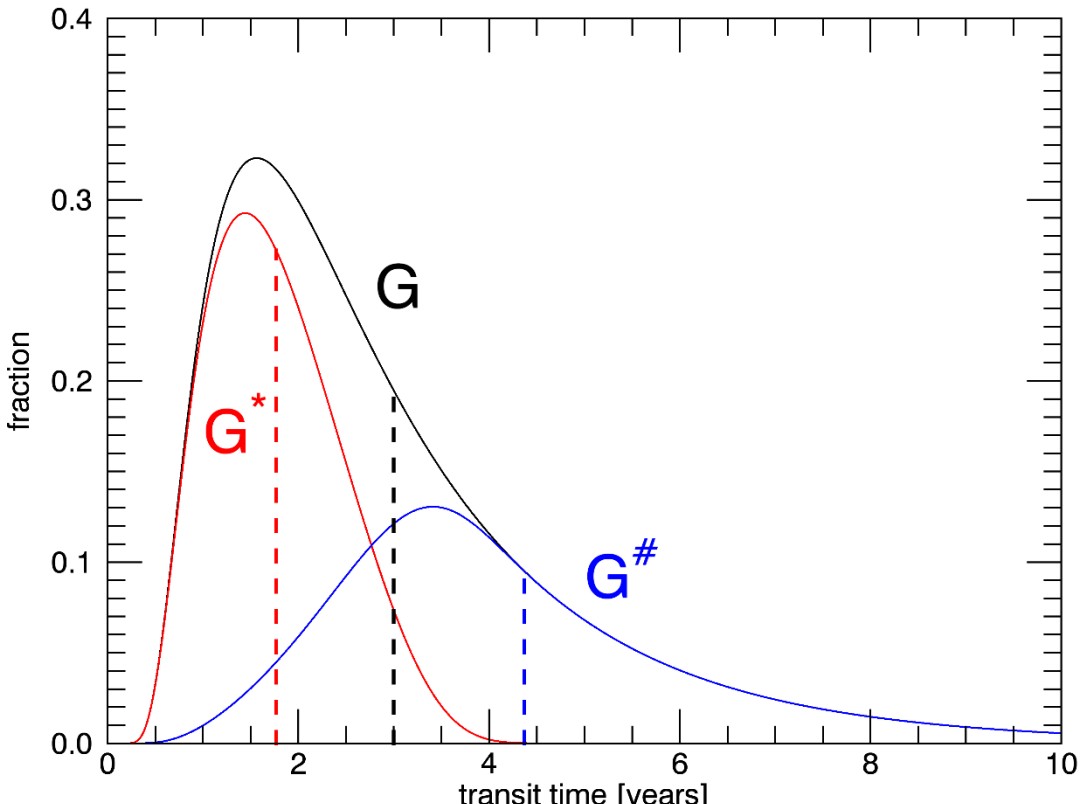

**Figure 1:** Age spectrum G (black line) for an inert tracer compared to the arrival time distribution $G^*$ (red line) and the release time distribution $G^\#$ (blue line). The loss function has been approximated as a function of transit time in order to represent a tracer similar to CFC-11 (see supplementary information for more details). The first moments of the three functions differ substantially: while the black line (inert tracer) has a first moment of 3 years, the first moment for the red curve, representing the remaining organic fraction is 1.75 years and that of the blue curve describing the inorganic halogen released from the source gas is 4.35 years. Note that these values are not identical to those for CFC-11 in Table A1 as the loss function was only approximated and that this Figure is purely for illustrative purposes.

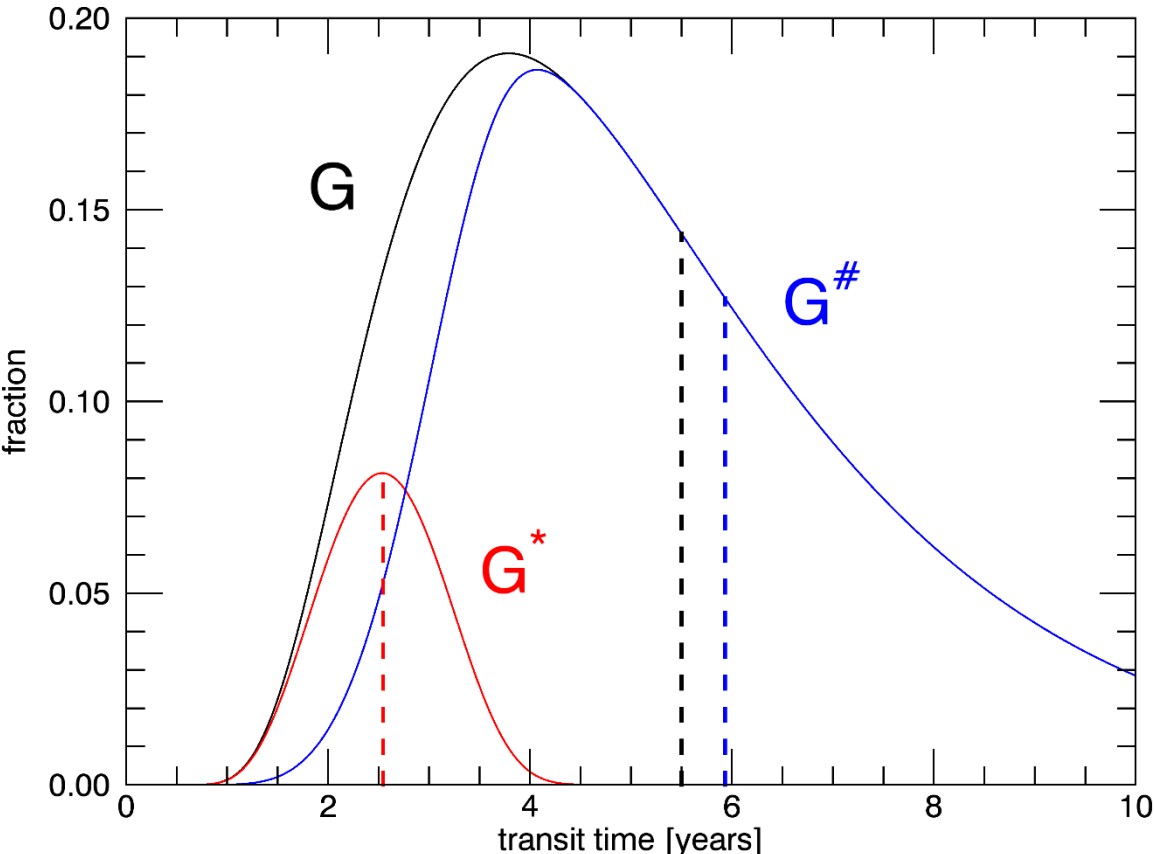

**Figure 2:** Age spectrum G (black line) for an inert tracer compared to the arrival time distribution G* (red line) and the release time distribution G# (blue line). The loss function has been approximated as a function of transit time in order to represent a tracer similar to CFC-11 (see supplementary information for more details). The first moments of the three functions differ substantially: while the black line (inert tracer) has a first moment of 5.5 years, the first moment for the red curve, representing the remaining organic fraction is 2.5 years and that of the blue curve describing the inorganic halogen released from the source gas is 5.9 years. Note that these values are not identical to those for CFC-11 in Table A2 as the loss function was only approximated and that this Figure is purely for illustrative purposes.

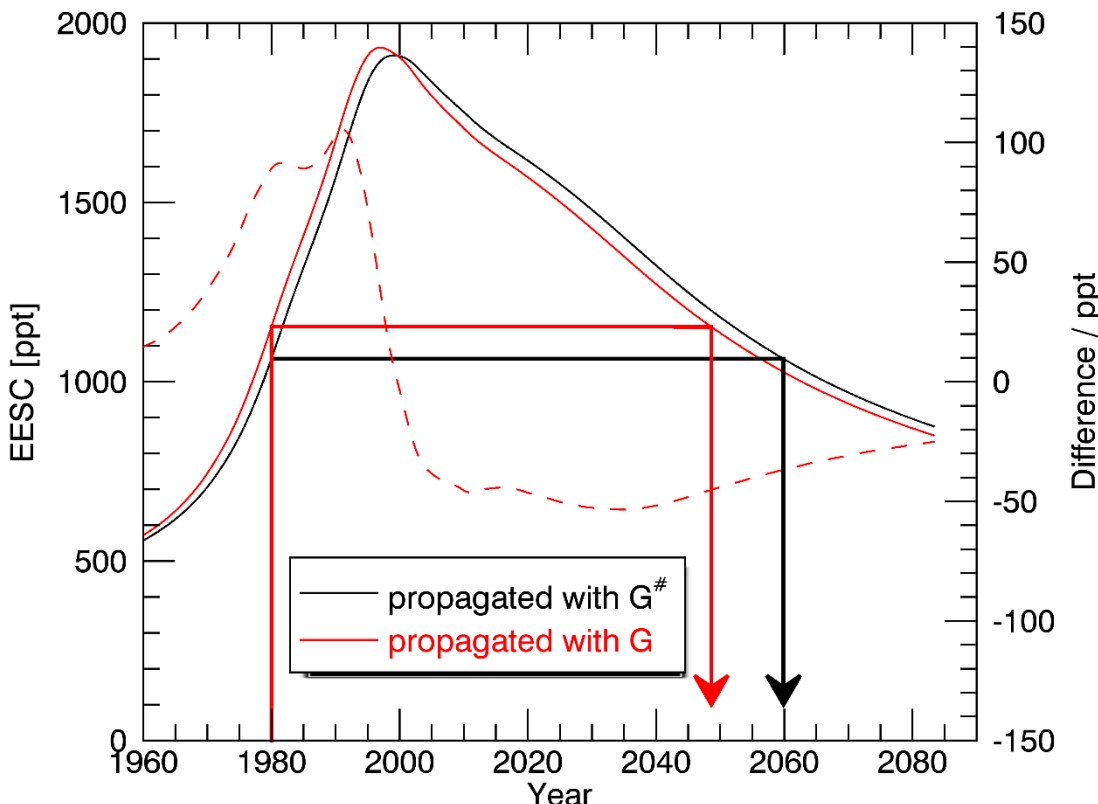

**Figure 3:** Estimated temporal evolution of EESC for a mean age of 3 years using the old (red line) and the new (black line) formulation of EESC. Also shown is the difference (red dashed line) and the recovery date to 1980 values for the old and the new formulation. Our new formulation yields a recovery date, which is more than 10 years later than using the current formulation. This shift in recovery date is mainly caused by the lower EESC levels calculated for the increasing phase, i.e. the 1980 reference value. All values given here are mole fractions given in ppt, which is equivalent to pmol/mol.

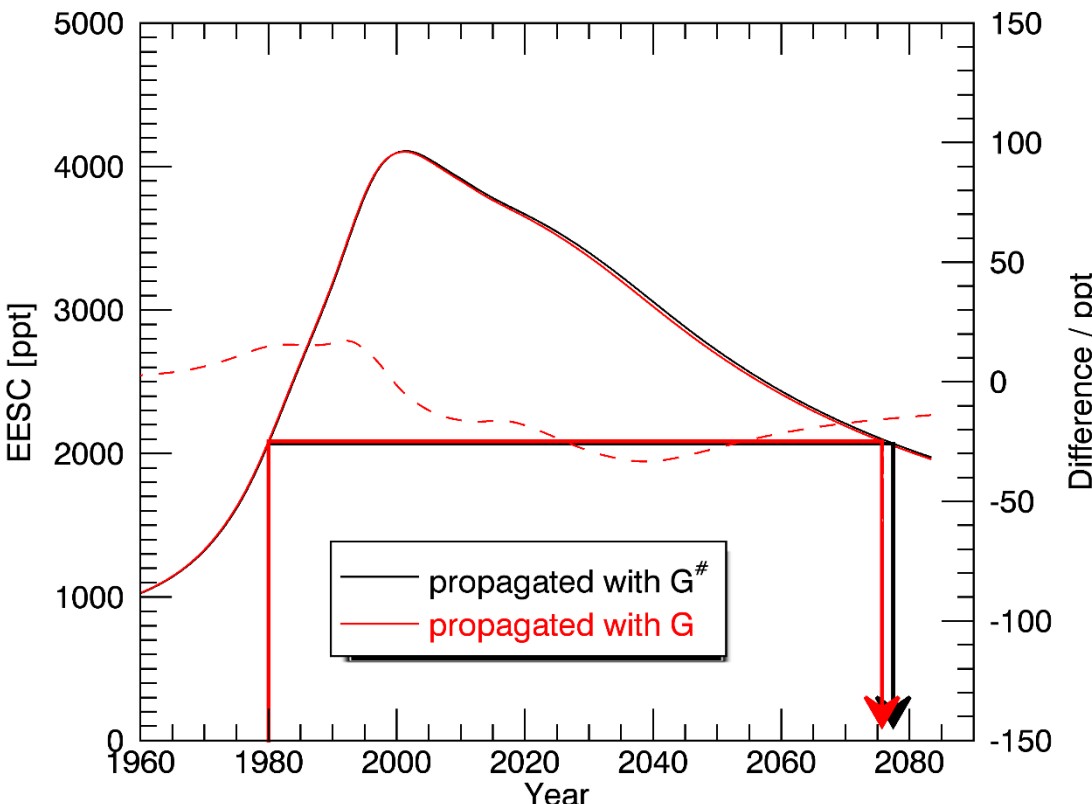

**Figure 4:** Estimated temporal evolution of EESC for a mean age of 5.5 years using the old (red line) and the new (black line) formulation of EESC. Also shown is the difference (red dashed line) and the recovery date to 1980 values for the old and the new formulation. Our new formulation yields a recovery date to 1980 values, which is about 2 years later than using the current formulation. The smaller shift in comparison to the calculation for 3 years of mean age is due to the near complete fractional release of most halogen source gases for these old air masses. All values given here are mole fractions given in ppt, which is equivalent to pmol/mol.

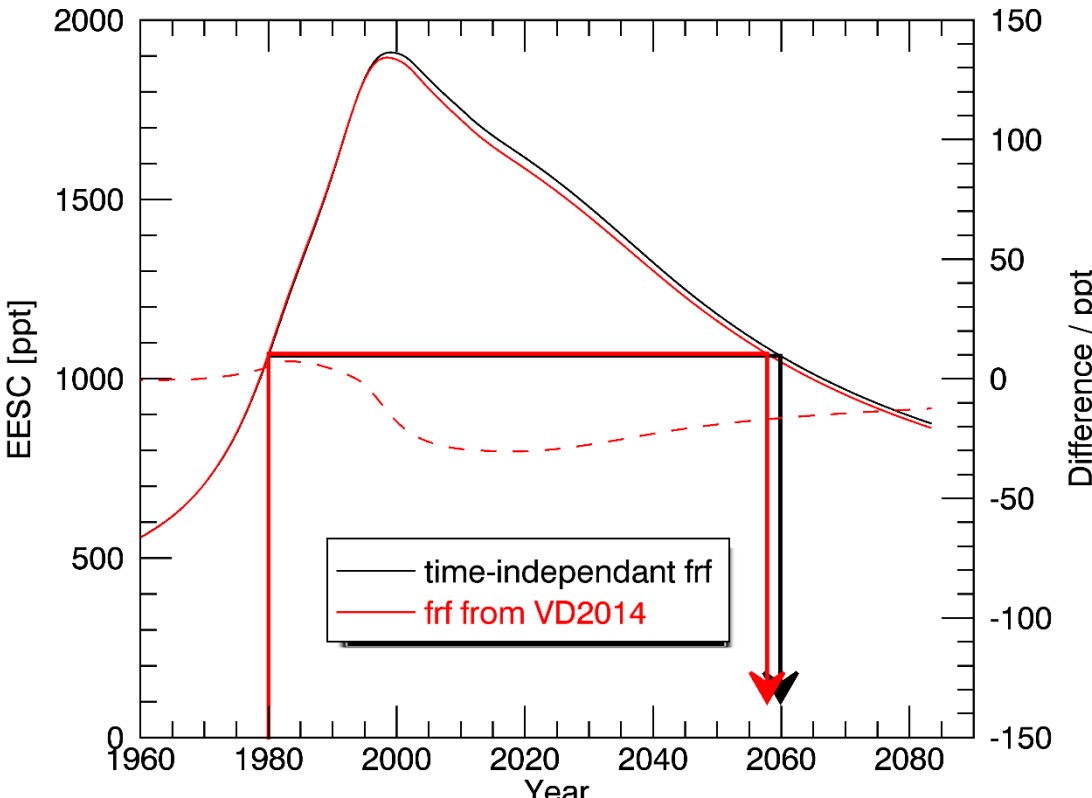

**Figure 5:** Influence of new fractional release factors for calculation of EESC at a mean age of 3 years. In both calculations the new formulation of EESC has been used, yet both calculations use different fractional release factors. The calculation using the new fractional release factors (Table A1) for 3 years of mean age is shown in black, while the calculation using the original values as used by Velders and Daniel (2014) (VD2014) is shown in red. As the fractional release factors currently used (Harris et al., 2014;Velders and Daniel, 2014) are largely based on measurements (Newman et al., 2007;Schauffler et al., 2003), which were taken during a period of rather small tropospheric trends for most species, the change due to the new formulation (Ostermöller et al., 2017) is rather small. For HCFCs 141b and 142b, uncertainties on observational fractional release factors are large and the same fractional release factors were used in both calculations, which are based on the parameterization given by Velders and Daniel (2014). The difference of the calculation using the VD2014 fractional release factors and our new time independent fractional release factors is shown as red dashed line. The fractional release factors used here are summarized in Table A1. All values given here are mole fractions given in ppt, which is equivalent to pmol/mol.

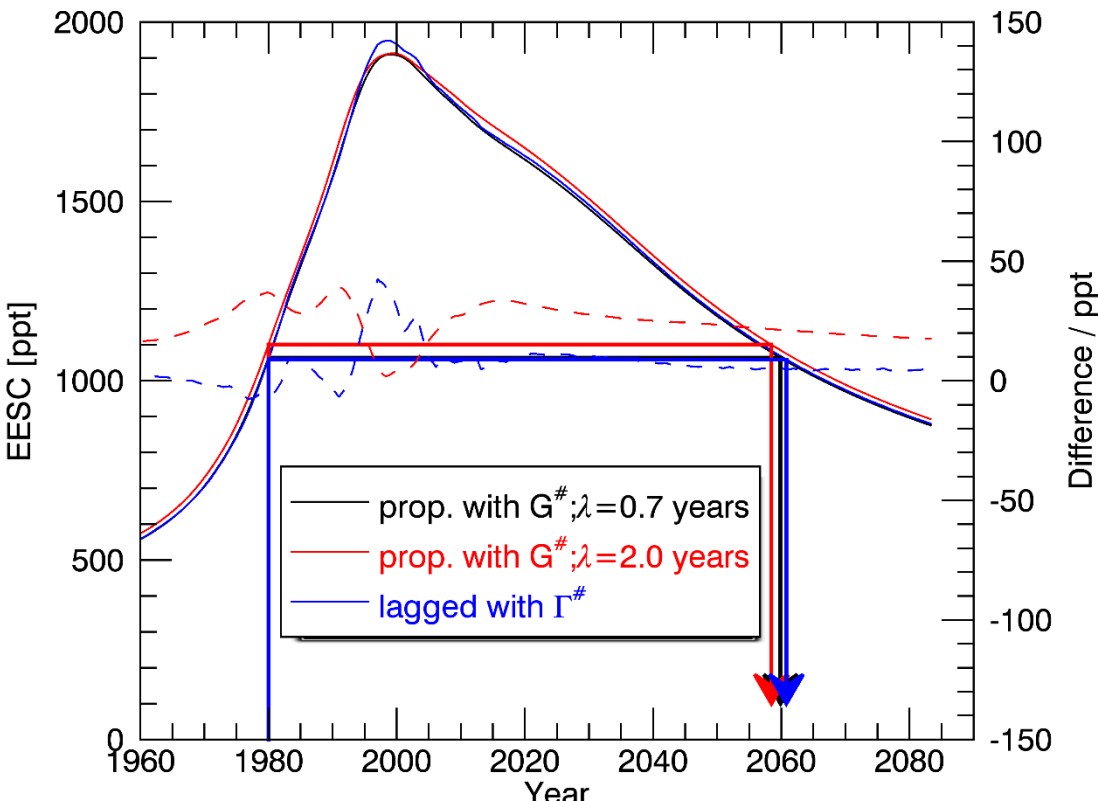

Figure 6: Sensitivity of EESC calculation using the new formulation for a mean age of 3 years on the parameterization of the shape of the release time distribution. In two cases the general shape was assumed to be an inverse Gaussian function with different parameterizations of the width $\Delta$, based on different values of $\lambda = \frac{\Delta^2}{\Gamma}$ (see text for an explanation of the parameterization). The case of the pure lag time calculation (blue line) is equal to $\lambda = 0$ years , i.e. the effect of mixing is completely ignored. The calculation using $\lambda = 2\ \text{years}$ (red line) represents a case with strong mixing and a broad transit time distribution, while $\lambda = 0.7\ \text{years}$ (black line) corresponds to our reference calculation. The influence is largest during the period of the maximum, when tropospheric trends showed a strong non-linear behavior and the tracer propagation strongly depends on the shape of the distribution function. The difference of the calculation to our reference calculations are shown as dashed lines. All values given here are mole fractions given in ppt, which is equivalent to pmol/mol.

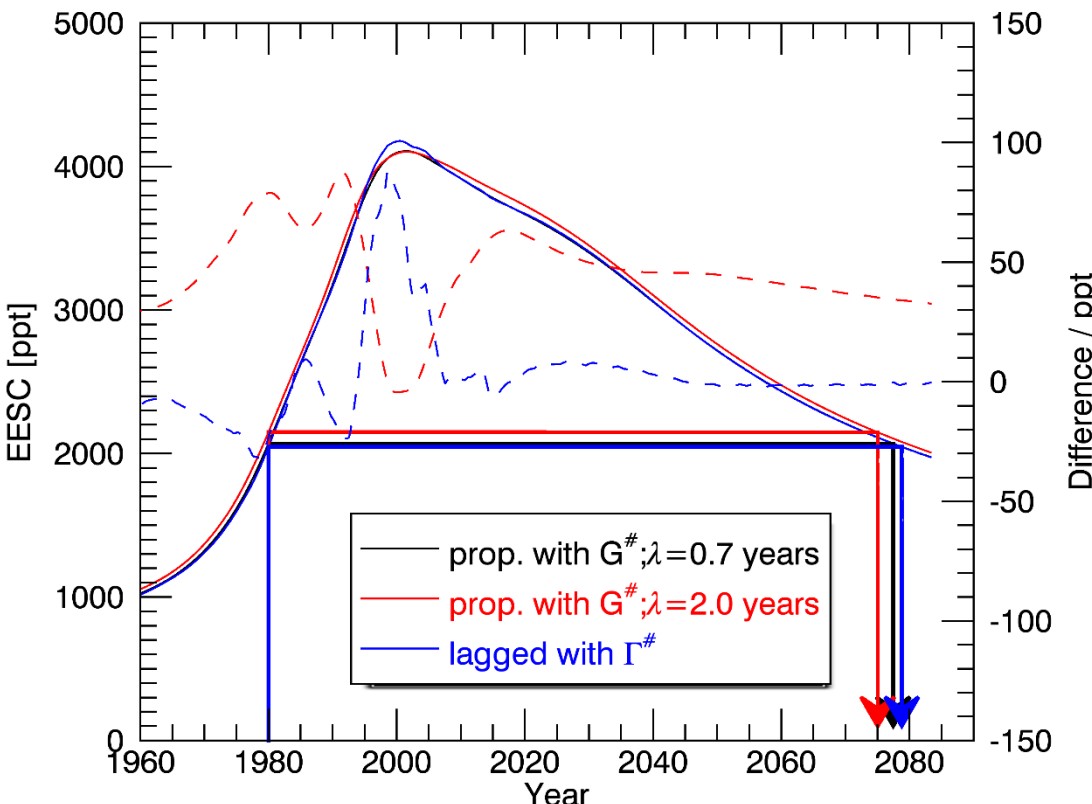

**Figure 7:** Sensitivity of EESC calculation using the new formulation for a mean age of 5.5 years on the parameterization of the shape of the release time distribution. In two cases the general shape was assumed to be an inverse Gaussian function with different parameterizations of the width $\Delta$, based on different values of $\lambda = \frac{\Delta^2}{\Gamma}$ (see text for an explanation of the parameterization). The case of the pure lag time calculation (blue line) is equal to $\lambda = 0$ years , i.e. the effect of mixing is completely ignored. The calculation using $\lambda = 2$ **years** (red line) represents a case with strong mixing and a broad transit time distribution, while $\lambda = 0.7$ **years** (black line) corresponds to our reference calculation. The influence is largest during the period of the maximum, when tropospheric trends showed a strong non-linear behavior and the tracer propagation strongly depends on the shape of the distribution function. The difference of the calculations to our reference calculation are shown as dashed lines. All values given here are mole fractions given in ppt, which is equivalent to pmol/mol.

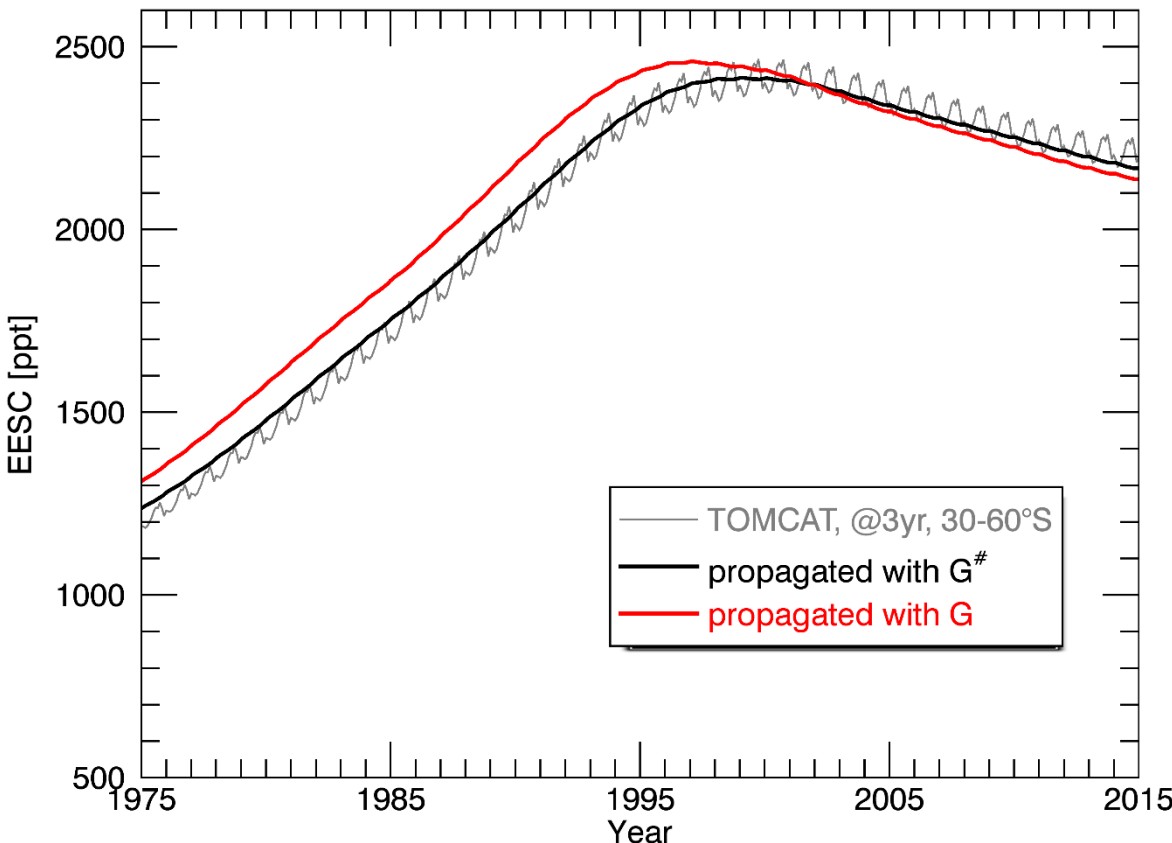

**Figure 8:** Comparison of EESC using the formulation by (Newman et al., 2007) and the new formulation suggested here to TOMCAT model calculations (Chipperfield et al., 2017) of ESC for Southern Hemisphere mid latitude conditions (3 years mean age). Fractional release values were calculated from the model and differ from those shown in Table 1, but are used in order for EESC and ESC to be consistent. The model simulation used here has fixed dynamics, using 1980 meteorology. While small differences remain, the new formulation yields much better agreement between EESC and ESC. All values given here are mole fractions given in ppt, which is equivalent to pmol/mol.

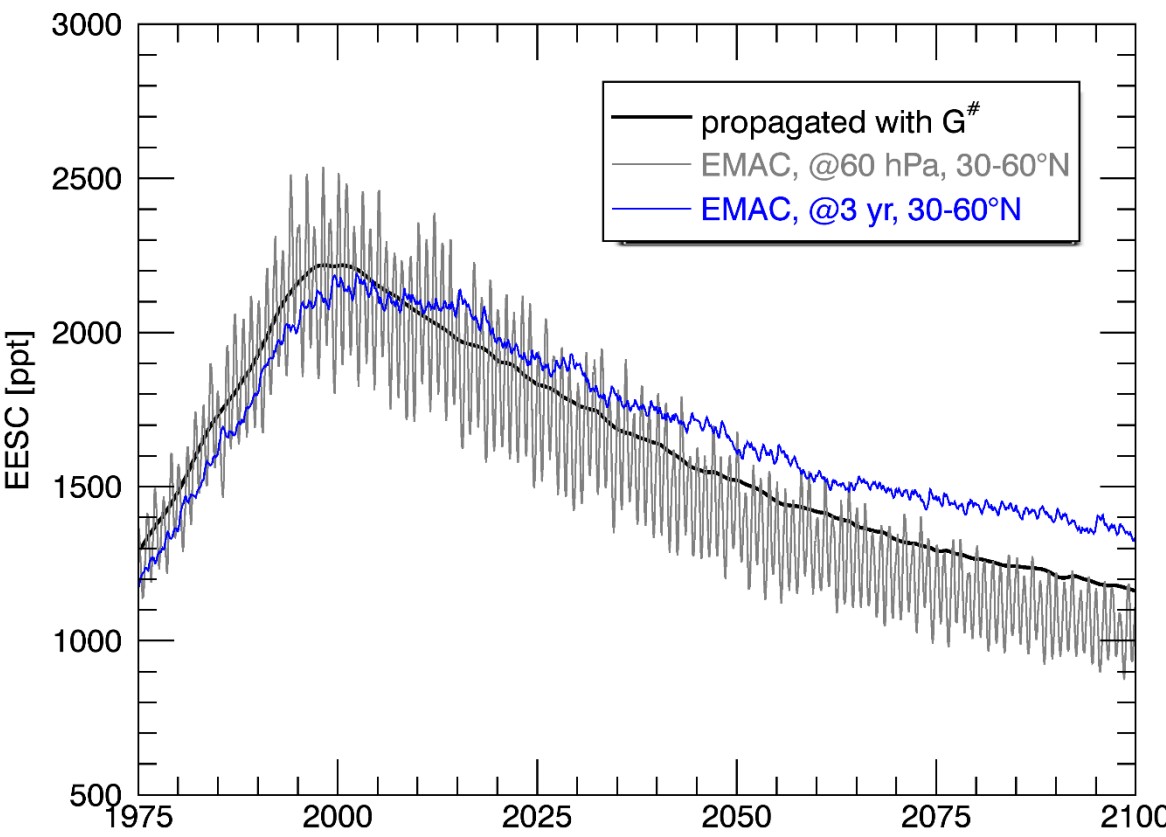

**Figure 9:** Comparison of EESC at 3 years of mean age using our new formulation to model ESC evaluated at the 60 hPa level (corresponding to 3 years of mean age in 2000) and to ESC at 3 years of mean age. Model data are from the EMAC model as described by Jöckel et al. (2016). Fractional release values were calculated from the model and differ from those shown in Table 1, but are used in order for EESC and ESC to be consistent. The model simulation shown here used prescribed trace gas scenarios, sea surface temperatures and sea ice content. All values given here are mole fractions given in ppt, which is equivalent to pmol/mol.