# Peer review of "A refined method for calculating Equivalent Effective Stratospheric Chlorine."

_Atmospheric Chemistry and Physics, 2017_

## Referee Comment (RC1) · Anonymous Referee #1 · 15 Aug 2017

Review of Delayed Recovery of mid-latitude lower stratospheric Halogen Loading By Engel, Bönich and Ostermöller

The authors lay a foundation for a new approach to computing EESC that attempts to account for some issues with this parameter. In addition, they draw the conclusion that recovery of mid-latitude lower stratospheric halogens (and ozone) will be delayed by 10 years compared with current projections.

I have many issues with this work. These start with the idea that any formulation of EESC is a certain metric for 'recovery'. It depends of course upon projections for many constituents, and therefore depends on their lifetimes and various other uncertainties. It gives a reasonable estimate for recovery, and a useful tool for quantifying the sensitivity of ozone to ODSs along with all of the other factors that influence ozone (solar,

volcanoes, qbo, etc.). I am generally skeptical that a change of a few percent in EESC changes anything about our understanding of recovery, or will make it easier to discern 'recovery' as time goes on. My concerns include some broad discussion points that I think can lead to misunderstanding as well as lack of referencing to a number of papers that are fundamental to the prior approaches to EESC calculation and use. There are also some overarching issues discussed below.

Overarching issues

1. There are different ways to think about the fractional release of ozone depleting substances. One way is to consider an age spectrum for elements, and to examine such factors as the maximum height achieved by parcels corresponding to each element. Statistical relationships can be obtained that make it possible to generalize the age spectrum to make sense for reactive gases. In older elements, most or all of the initial ODS has been destroyed. In younger elements, most or all of the initial ODS remains. The statistical relationship between the age and likelihood of destruction changes for source gases depending on their loss characteristics – specifically, how high must an element rise above the ozone peak in order to have experienced ODS destruction. Various papers take this approach – Hall (2000); Schoeberl et al. (2000); Schauffler et al. (2003); Schoeberl et al. (2005); Douglass et al. (2008). Distributions of various trace gases with a range of lifetimes show the validity of this approach as their lower stratospheric distributions carry information about the age spectrum. Waugh et al. (2007) show that the relationship of inorganic chlorine to mean age varies in different model implementations of the same meteorological fields, emphasizing the importance of a simulation reproducing not just the lower stratospheric mean age distribution but also distributions of various long-lived gases to attest to the realism of the age spectrum. It is especially important that appropriate relationships between maximum altitude and age are reproduced for all elements of the age spectrum. When this is achieved for a number of gases with varying lifetimes, then the relationship of fractional release as obtained from mean age, an observed gas, and the entry value to the stratosphere

estimated using the mean age makes sense as long as the dependence of the entry values is not a strong function of time.

These ideas become problematic if the sources are time dependent (and in particular when the sense of the time dependence changes from increasing to decreasing) especially when attempting to derive information from compact relationships that are obtained from observations for ODSs with a wide range of lifetimes. It may be necessary to account for the different values for entry to the stratosphere for various elements of the age spectrum rather than using the mean age to estimate that average. None of these things are well determined by observations, hence the reliance on some estimated width of the age spectrum. Many of these issues are less important in the current era, especially for the gases with lifetimes as long as CFCl3 and CF2Cl2, now that production is curtailed and the flux into the atmosphere from 'banks' is small compared to the present atmospheric burden.

The ideas presented above, where the contribution of each element in the age spectrum to an observed mixing ratio is controlled by its entry value, its path and the specifics of the loss for each ODS, are equivalent to the idea of 'arrival times' that vary for each ODS. Arrival times replaces one set of unknowns for another – by weighting the age spectrum with loss, you get a new 'time' quantity, different for each species, that controls the amount of a substance that you measure. This is a different twist on the same information that says that the long-lived elements control the amount that is destroyed, and applies a weighting function (different for each ODS) that is based on maximum altitude and probability of destruction to the same age spectrum . For a long-lived gas like CF2Cl2 more elements of the age spectrum contribute to the observed amount than for a shorter-lived element like CFCl3, thus more elements are in play in the distribution of arrival times. (Conversely, for a longer-lived gas fewer elements contribute to the observed inorganic chlorine etc.) From a physical point of view, I don't see the advantage of saying that the mean arrival time, shorter than the mean age, varies with species and that the age spectrum is 'inappropriate for a reactive gas' than
saying that the age spectrum is a statistical property of the flow, and that the amount of the gas observed is controlled by the statistical properties of the elements that produce the age spectrum and their ODS values at the time of entry to the stratosphere.

I think that to be convincing on the subject, the paper would have to explain this formulation in the context of work by the authors cited above and others published during the 2000s rather than relying on Plumb et al. 1999 which was primarily an attempt to generalize correlation relationships that depended on the time dependence of the source gases during a time when $CH_3CCl_3$ (methyl chloroform) was decreasing rapidly due to stoppage of production and its short life time.

2. The assumption underlying the projection on 'recovery' – that stratospheric dynamics and chemistry remain unchanged – is certainly false. The stratosphere is cooling and virtually all models project a speedup in the Brewer Dobson circulation. Although there is no consensus on details of how the circulation will change in the extratropics, and uncertainties in circulation change drive the differences in projections of the ozone layer for 2100, it is not sensible to make a projection that could only be true if the changes that are already observed to be occurring did not take place. The paper title asserting 'delayed recovery' is therefore misleading and unnecessarily alarming given the observational evidence that the column ozone loss at middle latitudes has been effectively limited ($\sim$4%, WMO 2010; WMO 2014) by the Montreal Protocol and its amendments.

3. The overall significance attached to EESC is unreasonably large. As the authors discuss, EESC is commonly used to separate ozone sensitivity to ODS build-up from other factors. The projection of delayed recovery comes from a lower value for EESC in 1980, thus return to 1980 will occur later. Does the new formulation change the ozone sensitivity that is obtained from the data record? Is there improvement in the statistical significance of the ozone sensitivity to chlorine change (e.g., is the overall fit to the prior record improved). Is there any observational evidence that shows that this formulation is both correct and beneficial? Does this formulation agree substan-

tially better with results from a 3D simulation (that passes the CCMVal transport tests) where the EESC can be calculated directly, where the mean age can be calculated either from pulsed experiments and these ideas can be fully tested, including physical importance? Finally, what would this mean with respect to the non-volcanic ozone loss already accrued by 1980 discussed by Shepherd et al. (2014)? Also note that Shepherd et al. compare the 'old' EESC with their calculation (summing up contributions to inorganic chlorine and bromine) and demonstrate that they agree.

Specific Major Comments

P 2 Line 5 Importance of EESC uses is overstated e.g., Tilmes et al. 2009 do not use EESC to discuss geoengineering. They compare their result to Newman et al. 2006, limited to polar ozone. Shepherd work shows that the conventionally derived EESC (Newman et al. 2007) agrees with the 3 D model result for the sum of inorganic chlorine and 60 times inorganic bromine . . . Weatherhead and Anderson (2006) use EESC as it is intended, as a parameter that generally describes the evolution of ozone depleting substances, but the subject of the paper is detectability of ozone trends and the discussion of the many factors that affect ozone, short term trends, and identification of 'recovery'.

An important question left undiscussed is whether and how this reformulation would change any of these results.

Page 3 Line 18 – EESC is 'influenced by . . . fractional release factors' This suggests that these factors are somehow invariant properties of the gases, whereas they are inherently tied to the meteorological factors that control the overturning circulation and mixing.

Page 4 and elsewhere: we discuss the interaction of chemistry and transport Chemistry and transport are not actually 'interacting' for these long-lived gases. The loss is a function of parcel path. The loss does not affect the parcel path, and therefore does not affect the transport. So what you are discussing is loss along a parcel path rather

than an interaction.

Page 4 section 2 Generally speaking, most of the loss takes place in the tropics (e.g., Prather et al., JGR 2015) N2O 81% between 24 and 40 km; 76% between 30S and 30N.

This section goes through a lot of details but the basic picture is not changed. A) the relationship of mean age, age spectrum, and the fractional release is not simple, as the contribution of each element to destruction of the ODS depends on parcel path. B) when comparing various species to their 'entry' values, in an ideal situation you have data on species of varying lifetimes so that you can obtain information about the mean age and the age spectrum from their differences; C) if the circulation is altered in any way such that the statistical relationships between age and path are altered for various elements of the age spectrum, then the empirical relationships will change. In fact, measurements of an inert gas such as SF6 in combination with measurements of various ODSs would provide information as to whether or not the age spectrum was changing.

Page 8 – the arrival time distribution and the release time distribution are 'closely linked'. This seems like it took a lot of work to get to a statement that follows directly from conservation. Also and probably more important – you don't ever cut to the chase for the simple physical principle. Lastly – the 'path independent fractional release' for a given mean age discussed in section 2 must have the relationship between loss, altitude and path imbedded in it – without some elements in the age spectrum reaching high altitude there will not be that much loss (at least for gases like CF2Cl2).

Section 4 In the end, you reach something that could be a simply stated conclusion. When EESC is growing rapidly, the relationship of mean age to older elements of the age spectrum is complicated because an entry value that is a simple function of the mean age does not take into account the very low entry values for the oldest elements of the spectrum. When EESC is decreasing, the reverse situation becomes the norm –

the old elements of the age spectrum have higher entry values than might be inferred from the mean age.

This work does not make a strong case for using the new formulation of age spectrum to project ozone recovery, rather it makes a case for uncertainty of EESC (although the values in Table 3 differ by less than 5% for mean age of 3 years), as well as a case for understanding the limitations of EESC. Given that midlatitude ozone loss is only $\sim$ 4%, it seems unlikely that the data up till now will differentiate one formulation from the other. It also seems unlikely that a difference in return to 1980 values would be discernable given all the other sources of variability (e.g., Mahieu et al., Nature, 2014) – the (difference between max and 1980)new – (difference between max and 1980)ccurrent is less than 10%.

Other Comments

Abstract: why 'can lead'? why not 'lead'?

Throughout: Please be careful and precise with 'depletion' and 'loss'. Loss is often natural. Depletion means loss greater than natural loss.

Conclusions: We suggest that this new method to calculate EESC be used to estimate the time of recovery of inorganic halogen to 1980 values – I suggest that you don't use EESC to estimate the time of recovery except in the broadest sense. Among other reasons – we don't have enough measurements of inorganic halogens in the 1980s to be confident of the starting value.

You propose comparing inorganic halogen levels from a full model calculation with the new EESC – I don't understand why you have not already used full model calculations (easily available through CCMI) to make this comparison.

Finally – if the new EESC formulation were shown to make a difference in the analysis of ozone time series, then that would be an important reason for its use. As it stands, this paper attempts to address a particular limitation to the current formulation for the

relationship of mean age, tracers, the age spectrum and fractional release. The discussion of the formalism does not make clear the reason for doing it clear from a physical point of view, and the geophysical significance of the result is overstated.
* * *

---

## Referee Comment (RC2) · Anonymous Referee #2 · 17 Aug 2017

This paper describes a reformulation of the EESC metric that attempts to account for the difference in stratospheric mass transport through photochemical loss regions vs. transport outside of the loss regions. The result is a relatively older mean age used to calculate EESC and thus a delay in future mid-latitude lower stratospheric EESC values to 1980 levels of roughly one decade.

Unfortunately, the paper has a number of fundamental flaws that make it unsuitable for publication. The major issues include (1) the absence of path height dependence in the fractional release and modified age distributions, (2) the assumption of constant stratospheric dynamics and photochemistry over nearly a century for the results to be valid, (3) the relatively minor role of EESC in future mid-latitude ozone depletion compared to N2O and (4) the absence of many relevant references and discussion of

previous work.

Main comments:

1. The authors mention in Section 2 that chemical loss is not uniform throughout the stratosphere and argue that on average the longer a fluid element remains in the stratosphere the larger the integrated chemical loss will be. This is not the case. The highly nonlinear loss dependence on altitude and latitude make the path an air parcel has taken much more important than the time an air parcel has resided in the stratosphere. Hall (2000) described the concept of maximum path height and the relationship between the path height and age distribution. Hall showed quite clearly that the mass fraction of an air parcel that has passed above some height, such as the height above which photochemical loss is rapid, is determined by the transport due to mass continuity and not the circulation, including mixing, that determines age distributions. Thus, the age distribution is only weakly linked to stratospheric photochemical destruction of any trace gas. This means that it is the circulation due to mass continuity that is most relevant in the estimation of EESC.

The circulation due to mass continuity is essentially the residual circulation and it has been shown, such as by Birner and Bonisch (2011) that the transit time due to the residual circulation is at most 3.5 years in the polar regions where the air parcels reached a minimum pressure of less than 0.1 hPa, well into the mesosphere, before descent into the polar vortices. This also implies that transit times throughout the mesosphere are actually less than 3.5 years. Mixing of air horizontally acts to increase the age of the stratosphere nearly everywhere (e.g. Garny et al., 2014) but not necessarily the maximum path height of a parcel and thus it's photochemical loss. The mixing is what drives the old tail in the age distribution and this mixing is not directly correlated with changes in path height.

The more appropriate time scale for photochemical loss is the mean arrival time at the location(s) where each trace gas is rapidly destroyed. Once an air parcel passes

through a region of rapid photochemical loss for a particular tracer the first time then all of that tracer is destroyed (converted into chlorine and bromine in this case) and it doesn't matter what happens to the air parcel from then on as far as the fractional release. Subsequent aging has no further effect on the release of chlorine or bromine. The mean arrival times at the region of rapid photochemical destruction for each trace gas will be dependent on the stratospheric transport each year and will be variable.

2. The assumption of constant stratospheric dynamics and photochemistry from the late 20th century to the late 21st century is certainly not a good one. What is the sensitivity of the results to the predicted changes in the stratospheric circulation?

3. $N2O$ is only mentioned briefly in the conclusions but mid-latitude ozone depletion in the late 21st century will be due primarily to $N2O$ concentrations (Ravishankara et al., 2009, Portmann et al., 2012, Butler et al., 2016). The variability and uncertainty in the $N2O$ concentrations will be much more of a factor in the return of mid-latitude ozone to 1980 levels than the small variability in the decline of EESC.

4. All of the above references are relevant to this study and none of them were included. The study of Waugh et al. (2007) is also highly relevant since they explore and discuss the age vs. path sensitivity of inorganic chlorine in the stratosphere.

A further point, what is special about recovery to 1980 levels? As was discussed in Newman et al. (2007) the year chosen to be the initial year causes large variability in the recovery time due to the steep slope of EESC around 1980 and the gradual slope in the late 21st century.

---

## Referee Comment (RC3) · Anonymous Referee #3 · 21 Aug 2017

The paper studies the combined effect of chemical loss and transport for the formulation of a simplified index describing the ozone depleting capacity due to halogens in the atmosphere. Usually, for that purpose the quantity EESC is used (for example in the WMO ozone assessments). EESC experienced several changes in its definition already in the past, when the correction for transport times was refined including its impact on release factors and the impact of transport time distribution when correcting for a non-linear trend has been included. Following Newman et al. (2007), the authors here present a further refinement of this concept.

Generally, I find this paper a very nice exercise to better understand how chemical loss and transport work together, here applied to describe the halogen loading in the atmosphere. The value of the paper is to remove inconsistencies in the formulation of

the traditional EESC concept, but still use the same simplified approach. The paper is clearly written, references are given as necessary, the assumptions are discussed in detail and possible consequences for the ozone recovery description are discussed.

My main obejection is the title of the paper which is in my opinion totally misleading. The paper is about a refinement of the EESC concept. In this concept, parameters can be derived like the time of recovery. This parameters has a descriptional character, and can be used to compare different scenarios of Halogen loading, for example. Modifiying the formulation of the concept will change the value of a parameter, but not the scenario, and therefore also not its "real" Halogen loading. In addition, as the authors state by themselves, the concept does not include deviations of the implicit assumed stationarity of the dynamics which is not even true for the past.

My second concern is somewhat related: as the EESC concept describes only one of the main driving processes for additional ozone depletion I would ask the authors therefore to put their results in the context of model studies where the effect of the accelerating Brewer Dobson circulation has been analysed.

The derivation of the concept is somewhat lengthy in my opinion and can be combined, for example for equs. (11)-(14). Stationarity means that at the end that the combined history of an air parcel (including its mean photochemical dose) is only a function of the position in the atmosphere. So equation (19) is not a surprise. Much more subtle is the transition from equ. (19) to (20). This is valid only under the specific condition that the distribution G# is determined by its first moment only which may be not generally true. Here I would ask the authors to discuss the assumptions in more detail.

Finally, to be more than just an interesting exercise, the paper would strongly improve if the authors could show that using their new formulation would yield a more concise ozone trend analyses, at least in one example.

Minor points

p2l9: transport within

p4l17: you mean averaged over the seasons == annual mean

p4l18: eliminate "it is expected"

p4l20: the typical path from the tropical tropopause through the stratosphere back to the troposphere at higher latitudes will not yield this shorter lifetime at the end of the path. You mean strictly in the stratosphere.

p4l21: f will be a function of r, too. This does not harm the derivation.

p5l31: what are the three? I see the trend and and the chemical loss only.

p6l2: the exponential chemical loss term is only used here and can be left out.

p11l9 remove "classical"

Typos:

parameterization/parameterization should be typed in one version only

p3l9: the first moment has a lower value

---

## Referee Comment (RC4) · Anonymous Referee #4 · 30 Aug 2017

This manuscript has built nicely on Ostermoller et al. (2017). The concept developed there is used to derive a relationship between previously calculated fractional release (FRF) values that assumed an age spectrum representative of an inert tracer to FRF values that are independent of tropospheric source gas trends. More importantly, this work quantifies the importance of using an age spectrum that accounts for chemical loss when calculating equivalent effective stratospheric chlorine (EESC). This improved approach effectively leads to older air in the EESC calculation, particularly for the mid-latitude stratosphere. This, in turn, implies lower EESC values in 1980; this 1980 level has been important because it has typically been taken as a value of significance in the return of stratospheric chlorine/bromine to natural levels. The proposed EESC revision (i.e., older air) also leads to higher EESC values for any given time when source gases

are declining. These changes combine to lead to a substantial delay in the time when mid-latitude EESC is projected to return to 1980 levels. As expected, the effects are smaller for polar EESC, since the difference in the average age for the dissociated ODSs and an inert tracer are much reduced.

I have a few general comments here, and some more specific ones below. Assuming these comments can be dealt with sufficiently, I find this manuscript to be valuable and I believe that it should offer an important improvement on work that came before it.

It would be useful to describe whether EESC from the new formalism is distinct enough from EESC using the old one so that past work that used EESC should have identified a shortcoming in the previous approach. Looking at Figure 3, I would be particularly interested in previous work that compared measurements or model calculations over a time range that spanned both before and after the EESC peak in the late 1990's, since the differences should be most apparent over such a period. If the two approaches are not distinct enough to be apparent in previous work, this would be worth stating here, so the reader knows the main impact is on the "recovery" date, and that it doesn't affect the validity of previous results.

The only other comment I particularly want to highlight here relates to the sensitivity study of the width of the transport distribution function. Please see my comment below for page 14, lines 2-3. I would find this most useful if you explored the impact of a change in width of the age distribution relevant to an inert tracer, with that impact propagating to the halocarbons depending on their chemical loss; however, unless I am mistaken, it doesn't seem like this is what is done.

Specific comments:

Page 1, Line 1 At some point, relatively early in the manuscript, you should make clear what you are not implying by this title, otherwise it could be considered misleading. As currently written, it could be taken to suggest that there has been more ODS emission than expected or that dynamics may change in an unexpected way to alter halogen

loading in the future. An alternative that may be preferable would be to change to a title more focused on the delay in EESC recovery.

1, 16 1980 is not the year of stratospheric ozone depletion onset, but it is often used as a benchmark to measure significant progress towards recovery

3,16 I suggest clarifying what 'this purpose' refers to at the end of this sentence

5,1-3 It is not clear to me that this sensitivity study addresses the entire phase space of possibilities in your assumed relationship between age and loss. Additional justification is needed to show that the simple relationship you are basing your calculations on are sufficiently appropriate.

6, 2 It is not clear to me that having the loss described as an exponential term with the lifetime depending on transit time is helpful in the formulation. It is really of an arbitrary mathematical form since the lifetime (denominator) varies with the location endpoint. It would seem more straightforward to skip straight to the factor $(1-f(t'))$, but I leave this decision to the authors.

10, 19 Somewhere you should discuss the impact of using the Plumb age estimates from an old 2-D model given the advancements in our ability to calculate circulation metrics over the last 20 years and the general superiority of 3-D models at making these calculations today

14, 2-3 I am having trouble understanding exactly what is being done here. Are the factors changed for both G and G#subN? It would not be possible to have gamma be 0 for G#subN and be 0.7 for G, would it? But if both gamma factors are 0, it would seem that this approach would collapse to the old result since the mean release age would be the same as the mean age of an inert species. And if that is the case, I would have expected a larger impact on return times (i.e., they should be close to the VD (2014) values). Perhaps it would help if you had a figure (like Figure 1) showing what the G curves look like as gamma goes to 0 and for it equal to 2. It looks like you may be

using the $\Gamma$# values from Table 1; however, this doesn't seem appropriate if you want to examine the impact of a changing shape in the overall transport distribution function. In fact, I'm unclear physically what is going on here, so clarification would be very helpful.

14, 25-29 Please describe how the destruction vs. age relationship is determined for these perturbations

15, 15 Perhaps broaden this statement, if you think it is accurate to something like "This approach more accurately represents the amount of Cly and Bry in the stratosphere from tropospheric source gas concentrations, and should be adopted to estimate..." This would seem to be more consistent with the title, but do please refer to my earlier concerns of such a broadening.

Minor comments:

1, 10 Do you mean 'adopted' here? While it is adapted through your work, that doesn't seem to be the intent here

1, 18 Change 'assumed' to 'estimated'

1, 27 Change to 'winter and springtime'

2,1 Replace 'effectiveness' with 'extent' or something similar; otherwise it could sound like the destruction per Cl molecule is what you are referring to here

2,2 Reference EESC

9, 1 Change 'was' to 'way'

9, 3 1-f(t) doesn't seem to be appropriately named as the 'loss term'; I understand why you called it the 'chemical loss term' back in eq. (6), but now in isolation is seems confusing that it is a 'loss term' that is really equal to one minus the fractional loss

12, 5 Change 'compares' to 'compare'

12, 27 Change 'on the new' to 'of the new' at the end of the line

13, 12 Add space in 'STRATcampaign'

13, 23-24 At first, this sentence seemed to suggest that you were doing another calculation from the Ostemoller results, but in fact, I believe you are summarizing the 2060 vs 2058 discussed at the top of this page. This could be clarified.

14, 18 Change to 'independent of'

15, 5 I don't understand the use of the word 'respective' here

15, 8 This seems to not be an appropriate use of 'Therefore'. From what is stated here, the following sentence doesn't seem to logically follow the previous statement(s)

15, 16/17 I suggest changing to present tense

15, 20 Change to '...perturbed values of stratospheric chlorine and bromine...'

15, 21-23 You should probably also point out that $CO_2$ is expected to accelerate column ozone recovery across much of the globe (see, e.g., Butler et al., 2016 and many others)

15, 33 It could be useful to say what G# is here, so people who didn't read the main text will know what it is.

16, 10 Make 'distributions' singular.

16, 17 Change 'calculation' to plural.

Table 1 For CFC-113 and $CH_3Br$ add another significant figure '0' to the end of the time-independent FRF. Same comment for various species in Table 2 in last 2 columns

21, 2 Perhaps add 'the' between 'In' and 'case'
* * *

---

## Author Comment (AC1) · 13 Nov 2017

**Answer to Reviewers**

We would like to thank all four reviewers for taking the time to go through the manuscript and giving us valuable feedback, which will help in improving the manuscript. Part of the criticism mainly from reviewer #1 and #2 is on the general use and concept of EESC and its use for the projection of a recovery date. This may be partly due to the choice of the title, which suggests a real change in recovery date. Actually the paper is mainly about the general concept of deriving EESC and therefore the title is indeed misleading. We suggest to change the title to "A refined method of calculating EESC" to emphasize that the idea is mainly to develop a concept which is mathematically consistent and physically more meaningful than the present formulation. We also understand the many limitations of the concept of EESC and that EESC will always be only a proxy for inorganic chlorine. Yet, EESC has some values, as it is a good proxy for one single parameter which has been subject to change due to a direct anthropogenic impact, namely the emission of halocarbons. It is therefore of considerable interest to understand, in particular in the context of the Montreal Protocol, when this impact is reversed and when the anthropogenic influence on this parameter will have reached a value which is deemed sufficiently close to background conditions. While the Brewer-Dobson circulation is expected to change in the future with considerable impact on inorganic chlorine in the stratosphere, it is exactly the idea behind EESC to have a measure of changes in this sole parameter of EESC. We also show in our revised manuscript that our new formulation is a reasonable proxy of inorganic halogen loading for the future on a given pressure level. We have added a new section (4.3.) to the manuscript which (i) compares our new formulation to that currently used based on a model calculation with fixed dynamics and (ii) shows that even under a changing circulation EESC is a reasonable proxy for inorganic halogen loading. Due to these additional comparisons with model calculations, we have added three additional co-authors to the paper, who provided model data for comparisons. We have also changed the abstract to reflect the changes. It does not contain a specific return data anymore but only mentions the time shift compared to the current formulation and also discusses the results from the model comparisons. New abstract:

*Abstract. Chlorine and bromine atoms lead to catalytic depletion of ozone in the stratosphere. Therefore the use and production of ozone depleting-substances (ODS) containing chlorine and bromine is regulated by the Montreal Protocol to*

*protect the ozone layer. Equivalent Effective Stratospheric Chlorine (EESC) has been adopted as an appropriate metric to describe the combined effects of chlorine and bromine released from halocarbons on stratospheric ozone. Here we revisit the concept of calculating EESC. We derive a refined formulation of EESC based on an advanced concept of ODS propagation into the stratosphere and reactive halogen release. A new transit time distribution is introduced in which the age spectrum for an inert tracer is weighted with the release function for inorganic halogen from the source gases. This distribution is termed*

*the "release time distribution". We show that a much better agreement to inorganic halogen loading from the chemistry transport model TOMCAT is achieved than using the current formulation. The refined formulation shows EESC levels in the year 1980 for the mid latitude lower stratosphere, which are significantly lower than previously calculated. 1980 marks the year commonly used as benchmark to which EESC must return in order to reach significant progress towards halogen and*

*ozone recovery. Assuming that – under otherwise unchanged conditions - the EESC value must return to the same level in order for ozone to fully recover, we show that it will take more than 10 years longer than estimated in this region of the stratosphere with the current method for calculation of EESC. We also present a range of sensitivity studies to investigate the effect of changes and uncertainties in the fractional release factors and in the assumptions on the shape of the release time*

*distributions. We further discuss the value of EESC as a proxy for future evolution of inorganic halogen loading under changing atmospheric dynamics using simulations from the EMAC model. We show that while the expected changes in stratospheric transport lead to significant differences between EESC and modelled inorganic halogen loading at constant mean age, EESC is a reasonable proxy for modelled inorganic halogen on a constant pressure level.*

One major fundamental comment to the paper, which was raised by reviewers 1 and 2 was that chemical breakdown of trace gases is not influenced by the transit time in the stratosphere, but that the location of transport pathway, and in particular the maximum path height (MPH) is the fundamental parameter, as discussed e.g. by (Hall, 2000) . This is in principle correct and we had partly addressed this in our paper, but probably not sufficiently. We had stated in the paper:

"*Chemical loss is not uniform throughout the stratosphere, as it depends in most cases on the actinic flux at short wavelength.*

*In the seasonal mean, the chemical lifetime will in general decrease with altitude and increase with increasing latitude. The chemical loss is thus very inhomogeneous, but on average, it is expected that the longer a fluid element remains in the stratosphere, the larger the integrated chemical loss will be. Also, as the loss for most species with photochemical sinks mainly occurs at higher altitudes, it is expected that, on average, longer transit times will be associated with shorter lifetimes.*"

Even better than the MPH, as stated in previous papers (Schoeberl et al., 2005;Hall, 2000), the total UV exposure along a transport path is the best measure for photochemical loss. This photochemical exposure as function of transit time is not available and will differ for different transit pathways even if they may have the same transit time.  Our approach relies on the assumption that **on average** there is a relationship between transit time and photochemical loss, so that on average the longer an air parcel has been in the stratosphere, the larger the chemical loss and thus the fractional release will be. Therefore, fractional release can be related to transit time. Figures 4, 6 and 8 in (Hall, 2000) actually show joint distributions showing that on average there is a correlation between transit time and MPH. This relationship between photochemical loss (for which MPH is also only a proxy) and transit times is actually the basis of the work of (Plumb et al., 1999) and (Vohralik et al., 1998) on which we base our concept and the parameterization of chemical loss. They were able to show that they could detrend correlations between different trace gases using the parameter of mean arrival time based on the concept that there is a clear relation between mean age and the photochemical degradation weighted mean arrival time. They state that "molecules arriving at X with long arrival times will, on average, have spent more time exposed to chemical loss and will have sampled atmospheric regions where photochemical loss is greater". This is exactly the prerequisite needed for our approach to describe the chemical loss as a function of transit time only without adding the path as an additional parameter. A detailed reply and how we addressed this concern is given in the individual reply to the reviewers (see foremost point 1 of reviewer 1).

In the following we will now answer the points raised by the reviewers point to point. As there are four reviews, we will partly refer to answers given to other reviewers. Our answers are shown in italic and changes to manuscript are shown in red.

**Reviewer 1**

Review of Delayed Recovery of mid-latitude lower stratospheric Halogen Loading By Engel, Bönisch and Ostermöller.
The authors lay a foundation for a new approach to computing EESC that attempts to account for some issues with this parameter. In addition, they draw the conclusion that recovery of mid-latitude lower stratospheric halogens (and ozone) will be delayed by 10 years compared with current projections.
I have many issues with this work. These start with the idea that any formulation of EESC is a certain metric for 'recovery'. It depends of course upon projections for many constituents, and therefore depends on their lifetimes and various other
uncertainties. It gives a reasonable estimate for recovery, and a useful tool for quantifying the sensitivity of ozone to ODSs along with all of the other factors that influence ozone (solar, volcanoes, QBO, etc.). I am generally skeptical that a change of a few percent in EESC changes anything about our understanding of recovery, or will make it easier to discern 'recovery' as time goes on. My concerns include some broad discussion points that I think can lead to misunderstanding as well as lack of referencing to a number of papers that are fundamental to the prior approaches to EESC calculation and use. There are also
some overarching issues discussed below.

*The reviewer is correct in pointing out that some reference to previous work was missing. We apologize for this and have now included these papers in the discussion (see details given below). We have further changed the title of the manuscript to "A refined method of calculating EESC" in order to lay the emphasis more on the principal method rather than on the issue of*
*recovery date. In this respect we would also like to point out that EESC is a parameter describing the evolution of one single parameter (inorganic halogen loading), which has been subject to change due to a direct anthropogenic impact in the past, namely the emission of halocarbons. It is therefore of considerable interest to understand, in particular in the context of the Montreal Protocol, when this impact is reversed and when the anthropogenic influence on this parameter will have reached a value which is deemed sufficiently close to background conditions. We have included the following statement on EESC at the*
*end of the first paragraph of the introduction:*

*Note that EESC is only a valid proxy for anthropogenic ozone depletion if all other parameters, especially atmospheric transport, are unchanged. A projection of a return of EESC to some specific level therefore does not imply that ozone will return to the same levels. EESC should thus be regarded as proxy for the impact of halogenated source gases on the ozone*
*layer due to both anthropogenic and natural emissions. The recovery of the ozone layer is affected by other parameters in addition, especially changes in transport. EESC is therefore not a proxy for ozone recovery, but a proxy for the impact due to one single parameter, the halogen loading. In section 4.3. we discuss the validity of EESC as a proxy for inorganic halogen loading under the influence of changing stratospheric dynamics.*

Overarching issues

1. There are different ways to think about the fractional release of ozone depleting substances. One way is to consider an age spectrum for elements, and to examine such factors as the maximum height achieved by parcels corresponding to each element. Statistical relationships can be obtained that make it possible to generalize the age spectrum to make sense for reactive gases.
In older elements, most or all of the initial ODS has been destroyed. In younger elements, most or all of the initial ODS remains. The statistical relationship between the age and likelihood of destruction changes for source gases depending on their loss characteristics – specifically, how high must an element rise above the ozone peak in order to have experienced ODS destruction. Various papers take this approach – Hall (2000); Schoeberl et al. (2000); Schauffler et al. (2003); Schoeberl et al. (2005); Douglass et al. (2008). Distributions of various trace gases with a range of lifetimes show the validity of this approach
as their lower stratospheric distributions carry information about the age spectrum. Waugh et al. (2007) show that the relationship of inorganic chlorine to mean age varies in different model implementations of the same meteorological fields, emphasizing the importance of a simulation reproducing not just the lower stratospheric mean age distribution but also distributions of various long-lived gases to attest to the realism of the age spectrum. It is especially important that appropriate relationships between maximum altitude and age are reproduced for all elements of the age spectrum. When this is achieved for a number of gases with varying lifetimes, then the relationship of fractional release as obtained from mean age, an observed gas, and the entry value to the stratosphere estimated using the mean age makes sense as long as the dependence of the entry values is not a strong function of time.

These ideas become problematic if the sources are time dependent (and in particular when the sense of the time dependence changes from increasing to decreasing) especially when attempting to derive information from compact relationships that are obtained from observations for ODSs with a wide range of lifetimes. It may be necessary to account for the different values for entry to the stratosphere for various elements of the age spectrum rather than using the mean age to estimate that average. None of these things are well determined by observations, hence the reliance on some estimated width of the age spectrum.
Many of these issues are less important in the current era, especially for the gases with lifetimes as long as CFCl3 and CF2Cl2, now that production is curtailed and the flux into the atmosphere from 'banks' is small compared to the present atmospheric burden.

The ideas presented above, where the contribution of each element in the age spectrum to an observed mixing ratio is controlled
by its entry value, its path and the specifics of the loss for each ODS, are equivalent to the idea of 'arrival times' that vary for each ODS. Arrival times replaces one set of unknowns for another – by weighting the age spectrum with loss, you get a new 'time' quantity, different for each species, that controls the amount of a substance that you measure. This is a different twist on the same information that says that the long-lived elements control the amount that is destroyed, and applies a weighting function (different for each ODS) that is based on maximum altitude and probability of destruction to the same age spectrum
. For a longlived gas like CF2Cl2 more elements of the age spectrum contribute to the observed  amount than for a shorter-lived element like CFCl3, thus more elements are in play in the distribution of arrival times. (Conversely, for a longer-lived gas fewer elements contribute to the observed inorganic chlorine etc.) From a physical point of view, I don't see the advantage of saying that the mean arrival time, shorter than the mean age, varies with species and that the age spectrum is 'inappropriate for a reactive gas' than saying that the age spectrum is a statistical property of the flow, and that the amount of the gas observed
is controlled by the statistical properties of the elements that produce the age spectrum and their ODS values at the time of entry to the stratosphere.

I think that to be convincing on the subject, the paper would have to explain this formulation in the context of work by the authors cited above and others published during the 2000s rather than relying on Plumb et al. 1999 which was primarily an
attempt to generalize correlation relationships that depended on the time dependence of the source
gases during a time when CH3CCl3 (methyl chloroform) was decreasing rapidly due to stoppage of production and its short life time.

*First we would like to point out that the maximum path height MPH is also only a proxy, as what is of interest is the cumulative*
*loss along a transit path, or as in our formulation for a transit time. The relation between transit time and chemical loss must only hold on average. There is no doubt that there will be parcels with long transit times but little loss. Nevertheless, on the average parcels with longer transit times do experience more loss, as also shown by e.g. (Plumb et al., 1999) on which a large part of work resides. This is also clearly visible in the work of (Hall, 2000) and the same approach has been used previously (Schoeberl et al., 2005;Schoeberl et al., 2000). This is the main point of criticism which we have addressed above. We think*
*that there is sufficient evidence also from the tightness of correlations between mean age and chemical active species, that on average there is a correlation between the transit time and chemical loss, which is the prerequisite for our approach to describe fractional release as a function of transit time only. Note that we do not need to make any assumptions on the exact shape of this relationship, as this is included in the release time distribution G#. In order to give credit to other work on this issue and to emphasize this point, we have changed the manuscript in section 2 as follows:*
*.*
*The fractional chemical loss can be expressed in a very generalized way as (1-f(t^',p)), where f(t^',p) is a fractional release function, which  is specific for each trace gas and will depend on the time the air parcel has spent in the stratosphere t' and on the path p it has taken during the transport, especially also of the maximum path height (MPH) (Hall, 2000). is While the path or the MPH are not known, it has been shown that "molecules arriving at X with long arrival times will, on average,*
*have spent more time exposed to chemical loss and will have sampled atmospheric regions where photochemical loss is*

*greater" (Plumb et al., 1999). We therefore follow the approach that the fractional loss can on average be described as function of the transit time only. That chemical loss and transit time are on the average related to each other is also reflected in the tight observed correlations between mean age and tracer mixing ratios (e.g.Volk et al., 1997;Engel et al., 2002). While there will be fluid elements with very different paths and different chemical loss which have the same transit time the loss can*

*on average be sufficiently well described as a function of the transit time. We therefore treat f(t^',p) as f(t^' ) only. This is also in line with the findings of Schoeberl et al. (2000), who showed that using an "average path approximation" with a "single-path photochemistry" and thus with a unique relationship between loss and transit time, global tracer-tracer correlations can be explained. This concept that loss can be described only as a function of transit time without considering the different transit pathways was also adopted by Schoeberl et al. (2005) in the derivation of age spectra. We use a mean age of 3 years for mid*

*latitudes and of 5.5. years for high latitudes. Indeed, the path distribution for an air parcel with mean age of 3 years in the tropics, in mid latitudes and in polar regions is expected to show more variability than for air parcels investigated under similar conditions (e.g.latitude regions). As this analysis is restricted to one latitude band for one mean age level, we therefore approximate loss as a function of transit time only.*

2. The assumption underlying the projection on 'recovery' – that stratospheric dynamics and chemistry remain unchanged – is certainly false. The stratosphere is cooling and virtually all models project a speedup in the Brewer Dobson circulation. Although there is no consensus on details of how the circulation will change in the extratropics, and uncertainties in circulation change drive the differences in projections of the ozone layer for 2100, it is not sensible to make a projection that could only be true if the changes that are already observed to be occurring did not take place. The paper title asserting 'delayed recovery'

is therefore misleading and unnecessarily alarming given the observational evidence that the column ozone loss at middle latitudes has been effectively limited (WMO 2010; WMO 2014) by the Montreal Protocol and its amendments.

*First of all, we have removed the term 'recovery' from the title of the manuscript, as it was indeed misleading. We have also emphasized in the introduction that EESC is a proxy for inorganic halogen loading only and not for the ozone layer and only*

*under the assumption of otherwise unchanged conditions. We have also added a new section 4.3. in which we compare our new calculation with the current formulation and model results, showing that it gives much better agreement and also comparing EESC to future inorganic halogen loading in a chemistry climate model with changing stratospheric transport. We have added the following in the introduction:*

*Note that EESC is only a valid proxy for anthropogenic ozone depletion if all other parameters, especially atmospheric transport, are unchanged. A projection of a return of EESC to some specific level therefore does not imply that ozone will return to the same levels. EESC should thus be regarded as proxy for the impact of halogenated source gases on the ozone layer due to both anthropogenic and natural emissions. The recovery of the ozone layer is affected by other parameters in addition, especially changes in transport. EESC is therefore not a proxy for ozone recovery, but a proxy for the impact due to*

*one single parameter, the halogen loading. In section 4.3. we discuss the validity of EESC as a proxy for inorganic halogen loading under the influence of changing stratospheric dynamics.*

*We have further added the section 4.3., here in particular the section comparing EESC to inorganic halogen in a full chemistry climate model. A reference to the sensitivity studies and model comparisons in section 4 has also been added at the end of the*

*introduction:*
*We also present sensitivity studies of the new EESC formulation to different parameters and compare the different formulations of EESC to simulations of inorganic halogen loading from two different comprehensive three-dimensional atmospheric chemistry models.*

**Comparison to model calculations with varying dynamics**

*Under changing stratospheric dynamics (e.g.Butchart, 2014), it is expected that fractional release factors at a given mean age level will change (Douglass et al., 2008;Li et al., 2012;Ostermöller et al., 2017). Therefore, the inorganic halogen loading as a function of mean age would be expected to change even if all source gases remained constant in time. Under such conditions, EESC is not expected to follow ESC on a given mean age level. To estimate the validity of EESC as a proxy for inorganic halogen loading of the stratosphere, we have compared our new formulation to a free running chemistry climate model simulation. We used data from the EMAC model simulation RC2-base-04 from the ESCiMo project for this (Jöckel et al., 2016). This simulation covers the 1950–2100 time frame with simulated sea surface temperatures and sea ice contents. As described above, we again calculated fractional release factors from the model in order to have results which are internally consistent. Northern Hemisphere fractional release factors for the year 2000 are in good agreement with observation based fractional release factors (Newman et al., 2007;Laube et al., 2013) and therefore we used northern hemispheric data for this comparison. In addition to comparing ESC on a fixed mean age level we also compared ESC on a fixed pressure level to our new formulation of EESC. A similar comparison has been presented in Shepherd et al. (2014), who compared model ESC on a fixed pressure level to EESC on fixed mean age level (using the formulation of Newman et al. (2007)), showing good agreement. Figure 9 compares the time evolution of EESC for 3 years of mean age with model ESC at the 60 hPa level (corresponding to 3 years of mean age in the year 2000) and model ESC at 3 years of mean age. The year of 2000 and the corresponding level of 60 hPa was chosen, as we also evaluated fractional release factors in the year 2000 of the model run. As expected, ESC at 3 years of mean age deviates systematically from EESC, especially in the future when fractional release evaluated on a mean age surface changes significantly in the model. The agreement with ESC on a fixed pressure level is however much better. In this comparison EESC at 3 years of mean age would slightly overestimate ESC on a pressure level in the future and significantly underestimate ESC on a mean age level. The exact magnitude of changes in stratospheric dynamics is highly uncertain and also we expect ozone to follow a pressure surface rather than a mean age surface in the future. We therefore conclude that EESC is a reasonable proxy for the effect of halogen loading on stratospheric ozone, given the overall high uncertainties associated to the future evolution of stratospheric dynamics.*

*Further we have added in the discussion section:*

*We have shown that the long-term evolution of effective stratospheric chlorine (ESC, i.e. inorganic chlorine and bromine, the latter weighted in a similar way as in EESC to reflect the higher efficiency of bromine to ozone depletion) in the model deviates substantially from our calculation of EESC in a long-term model calculation with varying dynamics. However, we have also shown that the new formulation of EESC is a reasonable proxy for the evolution of inorganic halogen loading on a given pressure level. We therefore conclude that EESC is a reasonable proxy for future halogen impact on ozone.*

*...*

*As current climate models consistently predict an acceleration in the Brewer-Dobson circulation (Butchart, 2014), this will have an impact on the temporal evolution of inorganic halogen loading of the stratosphere. These expected changes in the Brewer-Dobson circulation would result in an earlier recovery of ozone at mid- and high latitudes (Eyring et al., 2010). These*

*changes are not included in the concept of EESC. However, we have shown that EESC is a reasonable proxy for ESC when ESC is evaluated at constant pressure level.*

3. The overall significance attached to EESC is unreasonably large. As the authors discuss, EESC is commonly used to separate ozone sensitivity to ODS build-up from other factors. The projection of delayed recovery comes from a lower value for EESC in 1980, thus return to 1980 will occur later. Does the new formulation change the ozone sensitivity that is obtained from the data record? Is there improvement in the statistical significance of the ozone sensitivity to chlorine change (e.g., is the overall fit to the prior record improved). Is there any observational evidence that shows that this formulation is both correct and
beneficial? Does this formulation agree substantially better with results from a 3D simulation (that passes the CCMVal transport tests) where the EESC can be calculated directly, where the mean age can be calculated either from pulsed experiments and these ideas can be fully tested, including physical importance? Finally, what would this mean with respect to the non-volcanic ozone loss already accrued by 1980 discussed by Shepherd et al. (2014)? Also note that Shepherd et al. compare the 'old' EESC with their calculation (summing up contributions to inorganic chlorine and bromine) and demonstrate
that they agree.

*In the model inorganic halogen (ESC, Equivalent stratospheric chlorine) can be directly determined from Cly and Bry. Even if there were no trends at all in organic halogen source gases, ESC on a given mean age level would change in the model, as fractional release is expected to change on a given mean age level. Interestingly, this change in fractional release is less*
*pronounced on pressure levels. (Shepherd et al., 2014) compared the evolution of ESC at 50 hPa with EESC at 3 and 5 years of mean age. As the mean age on a given pressure level changes, these are to a certain degree two different things which are compared with each other, showing surprisingly good agreement.*

*A meaningful comparison of ESC and EESC is far from trivial. We have compared our new and the old formulation of EESC*
*to an EMAC simulation (Jöckel et al., 2016) and found that the ESC in the model deviates substantially from EESC (both formulations), mainly due to long term changes of frf on mean age levels. As in (Shepherd et al., 2014), we find a much better agreement when comparing EESC on a mean age level with ESC on a given pressure level. On the other side, the mean age on the pressure level changes, so this comparison is not really valid. A more meaningful comparison between ESC and EESC can be done using a model run with fixed dynamics and changing ODS levels. We now show a comparison between our new*
*formulation and a model run from the TOMCAT model (Chipperfield et al., 2017), which uses fixed dynamics for the year 1980. In this model run, a meaningful comparison can be performed and we show that there is much better agreement between our new formulation and the model calculation when compared to the old formulation. We have added a new section to the manuscript (section 4.3), called "Comparison of EESC formulations with model calculations of inorganic halogen loading" in which we show the comparison with the EMAC model (with changing dynamics) and the TOMCAT model (with fixed*
*dynamics). Section 4.3. on comparison with fixed dynamics calculations:*

**4.3. Comparison of EESC formulations with model calculations of inorganic halogen loading.**

*In order to evaluate our new formulation of EESC we have compared the results of our calculations with the inorganic halogen loading calculated from two comprehensive three-dimensional atmospheric chemistry models. Due to expected long-term changes in mean age on a given pressure level associated with the simulated changes in the Brewer-Dobson circulation (e.g.*
*Butchart, 2014;Austin and Li, 2006), changes in fractional release factors on mean age levels are also observed in free running model calculations (Douglass et al., 2008;Li et al., 2012). We have therefore compared our new formulation of EESC to model calculations with changing and with annually repeating ('fixed') dynamics. To compare the new formulation with the formulation by (Newman et al., 2007), we used a model simulation from the TOMCAT model (Chipperfield et al., 2017), which was driven by a repeated meteorology, in this case for the year 1980. Effects due to changing dynamics, which are not included*

*in the concept of EESC, will thus not impact this calculation, making it an ideal test bed for comparison of the two formulations. For long term changes, we have used model results from the EMAC model (Jöckel et al., 2016), which includes expected changes in stratospheric transport. As in general the relationship between mean age and $Cl_y$ is very different for different models (Waugh et al., 2007), a direct comparison between EESC and ESC (Equivalent stratospheric Chlorine, calculated from*

*model $Cl_y$ and $Br_y$ using the same sensitivity parameters for bromine as with EESC) is not meaningful, as differences may be due to different fractional release factors between models and observations. Instead we have used fractional release values derived from the models for 3 years of mean age and used these for the calculation of EESC using the formulations by Newman et al. (2007) and from this work. Fractional release factors were calculated from the model data using the methods of Newman et al. (2007) and Ostermöller et al. (2017) for this work. The fractional release factors were calculated for the year 2000, in*

*order to be consistent with the observation based fractional release factors, which were derived mainly for the year 2000. To the EESC calculated in this way we added simulated inorganic chlorine and bromine at the tropical tropopause, which we propagated as an inert tracer. VSLS (very short lived substances) were treated in a similar way, using their tropical tropopause values as input, as loss in the troposphere cannot be neglected for these species. As no global stratospheric lifetimes are available for these species, it is not possible to apply the new formulation. Therefore, the VSLS were treated using the method*

*of Newman et al. (2007). The differences are negligible, as the VSLS have rather slow long-term trends and both methods yield nearly identical results, as also discussed in Ostermöller et al. (2017). As some loss of $CH_3Cl$ and $CH_3Br$ occurs during the transport in the troposphere to the tropical tropopause, we have also used the time series of these two gases at the tropical tropopause rather than at the surface in these calculations.*

**Comparison to fixed dynamics model calculations**

*For comparison of the two formulations to model calculations with fixed dynamics, we used a TOMCAT model run (Chipperfield et al., 2017), which is driven by repeated 1980 meteorology. This model run is available from 1960 through to 2016. The fractional release factors derived from the model for the northern hemisphere are significantly higher as function of mean age than the observed fractional release values. Southern hemispheric fractional release values for 3 years of mean age showed better agreement with observation derived fractional release factors (Newman et al., 2007). For this reason we*

*compared simulated ESC from the southern hemisphere with EESC calculated using our new formulation and the formulation by (Newman et al., 2007), in both cases using fractional release values derived for the year 2000 model results.*

*Figure 8 shows the comparison between the modelled ESC and EESC for a mean age of 3 years calculated as described above, including all bromine and chlorine species included in the model and also including inorganic chlorine and bromine entering the stratosphere. As the differences between the two formulations of EESC are most pronounced for 3 years of mean age, we*

*show this comparison for 3 years of mean age only. A much better agreement is observed when applying the new formulation than using the formulation by Newman et al. (2007), due to the improved treatment of the combined influence of transport and*

*mixing on chemical loss. Remaining discrepancies between model ESC and EESC are most probably due to an imperfect parameterization of the loss time distribution $G^{\#}$.*

Specific Major Comments

P 2 Line 5 Importance of EESC uses is overstated e.g., Tilmes et al. 2009 do not use EESC to discuss geoengineering. They compare their result to Newman et al. 2006, limited to polar ozone. Shepherd work shows that the conventionally derived EESC (Newman et al. 2007) agrees with the 3 D model result for the sum of inorganic chlorine and 60 times inorganic bromine . . . Weatherhead and Anderson (2006) use EESC as it is intended, as a parameter that generally describes the evolution of ozone depleting substances, but the subject of the paper is detectability of ozone trends and the discussion of the many factors that affect ozone, short term trends, and identification of 'recovery'.

An important question left undiscussed is whether and how this reformulation would change any of these results.

*This is a good point. We do not think that the reformulation would affect the results of these studies in a qualitative way. It is*
*only when EESC is used in a quantitative way, e.g. in the projection of possible changes using different scenarios and only when frf values are significantly lower than 1 that this reformulation is expected to have a significant impact. We have added the following sentence in the conclusions:*

*As the suggested reformulation of EESC does not affect the principal behavior of the temporal evolution of EESC, we do not*
*expect this reformulation to lead to substantial changes, which could impact the changes of studies using EESC except for those which have used EESC to project EESC recovery.*

Page 3 Line 18 – EESC is 'influenced by . . . fractional release factors' This suggests that these factors are somehow invariant properties of the gases, whereas they are inherently tied to the meteorological factors that control the overturning circulation
and mixing.

*This was certainly not the intention of that statement. But fractional release factors should be independent of tropospheric trends of ODS. In order to make this clearer we have added the following statement behind this sentence:*

*While all these factors may vary with time, e.g. due to changes in stratospheric circulation (Douglass et al., 2008;Li et al., 2012), especially fractional release factors should not depend on the tropospheric trends of the trace gases (Ostermöller et al., 2017).*

Page 4 and elsewhere: we discuss the interaction of chemistry and transport. Chemistry and transport are not actually
'interacting' for these long-lived gases. The loss is a function of parcel path. The loss does not affect the parcel path, and therefore does not affect the transport. So what you are discussing is loss along a parcel path rather than an interaction.

*We only partly agree. Of course the chemistry has a feed-back on the transport via radiation and heating rates. Nevertheless we suggest to rephrase the title of this section to "On the influence of transport and tropospheric trends on chemical active*
*species".*

Page 4 section 2 Generally speaking, most of the loss takes place in the tropics (e.g., Prather et al., JGR 2015) N2O 81% between 24 and 40 km; 76% between 30S and 30N. This section goes through a lot of details but the basic picture is not changed. A) the relationship of mean age, age spectrum, and the fractional release is not simple, as the contribution of each
element to destruction of the ODS depends on parcel path. B) when comparing various species to their 'entry' values, in an ideal situation you have data on species of varying lifetimes so that you can obtain information about the mean age and the age spectrum from their differences; C) if the circulation is altered in any way such that the statistical relationships between age and path are altered for various elements of the age spectrum, then the empirical relationships will change. In fact, measurements of an inert gas such as SF6 in combination with measurements of various ODSs would provide information as to whether or not the age spectrum was changing.

*We are not sure, what the reviewer is suggesting here. In principle we agree to the statements, but we would like to emphasize again, that the current formulation of EESC does not take into account that fractional release is in any way dependent on the transport path or transit time. In our new formulation this dependency is added, by using a transit time dependent approach for fractional release (and assuming that on average longer transit times are related to more chemical loss). Regarding the possibility of deriving age spectra and changes of age spectra from observations, this is a very different subject and not the scope of this paper. And indeed, fractional release as a function of mean age is expected to change. This is, however, not included in EESC, neither in the old nor in our new formulation.*

Page 8 – the arrival time distribution and the release time distribution are 'closely linked'. This seems like it took a lot of work to get to a statement that follows directly from conservation. Also and probably more important – you don't ever cut to the chase for the simple physical principle. Lastly – the 'path independent fractional release' for a given mean age discussed in section 2 must have the relationship between loss, altitude and path imbedded in it – without some elements in the age spectrum reaching high altitude there will not be that much loss (at least for gases like CF2Cl2).

*This touches again on the main point addressed at the top of our reply. We do not suggest that fractional release is path independent but rather we suggest that this path dependence can be treated as a transit time dependency given that, on average, longer transit times are associated with more time spent in the chemical loss region in the upper tropical stratosphere and thus more chemical loss.*

Section 4 In the end, you reach something that could be a simply stated conclusion. When EESC is growing rapidly, the relationship of mean age to older elements of the age spectrum is complicated because an entry value that is a simple function of the mean age does not take into account the very low entry values for the oldest elements of the spectrum. When EESC is decreasing, the reverse situation becomes the norm – the old elements of the age spectrum have higher entry values than might be inferred from the mean age.

This work does not make a strong case for using the new formulation of age spectrum to project ozone recovery, rather it makes a case for uncertainty of EESC (although the values in Table 3 differ by less than 5% for mean age of 3 years), as well as a case for understanding the limitations of EESC. Given that midlatitude ozone loss is only _ 4%, it seems unlikely that the data up till now will differentiate one formulation from the other. It also seems unlikely that a difference in return to 1980 values would be discernable given all the other sources of variability (e.g., Mahieu et al., Nature, 2014) – the (difference between max and 1980)new – (difference between max and 1980) current is less than 10%.

*In order to avoid a wrong impression that might have be caused by the title, we have changed the title, to emphasize that the main purpose of this paper is a new, mathematical consistent, formulation of EESC. In the same time, we have slightly different understanding of what EESC should provide, as its main purpose is to look at the halogen-induced impact on the stratosphere, where it can be used as proxy and also as metric to assess when this effect is reduced to levels which are comparable to those present in the stratosphere at some reference time set (rather arbitrarily) to 1980. We have made this clear by including the following statement in the introduction:*

*Note that EESC is only a valid proxy for anthropogenic ozone depletion if all other parameters, especially atmospheric transport, are unchanged. A projection of a return of EESC to some specific level therefore does not imply that ozone will return to the same levels. EESC should thus be regarded as proxy for the impact of halogenated source gases on the ozone layer due to both anthropogenic and natural emissions. The recovery of the ozone layer is affected by other parameters in addition, especially changes in transport. EESC is therefore not a proxy for ozone recovery, but a proxy for the impact due to one single parameter, the halogen loading. In section 4.3. we discuss the validity of EESC as a proxy for inorganic halogen loading under the influence of changing stratospheric dynamics.*

*We have further added the section 4.3. which describes the comparison of our new EESC formulation and the formulation by Newman to model calculations.*

Other Comments

Abstract: why 'can lead'? why not 'lead'? *Has been changed.*

Throughout: Please be careful and precise with 'depletion' and 'loss'. Loss is often natural. Depletion means loss greater than natural loss.

*Thank you, we have made sure that depletion is only for anthropogenic and loss for natural and anthropogenic loss. In most cases loss is used with respect to ODS and not with respect to ozone. .*

Conclusions: We suggest that this new method to calculate EESC be used to estimate the time of recovery of inorganic halogen to 1980 values – I suggest that you don't use EESC to estimate the time of recovery except in the broadest sense. Among other reasons – we don't have enough measurements of inorganic halogens in the 1980s to be confident of the starting value. You propose comparing inorganic halogen levels from a full model calculation with the new EESC – I don't understand why you have not already used full model calculations (easily available through CCMI) to make this comparison. Finally – if the new EESC formulation were shown to make a difference in the analysis of ozone time series, then that would be an important reason for its use. As it stands, this paper attempts to address a particular limitation to the current formulation for the relationship of mean age, tracers, the age spectrum and fractional release. The discussion of the formalism does not make clear the reason for doing it clear from a physical point of view, and the geophysical significance of the result is overstated.

*We only partly agree to this statement that EESC should not be used to project a recovery. EESC has been used regularly in the WMO ozone assessment reports to estimate the level of recovery already achieved due to the regulations of the Montreal Protocol and also to project future halogen loading of the stratosphere and possible recovery dates for this halogen loading to the admittingly partly arbitrary benchmark level of 1980. EESC certainly has limitations, but on the other hand it has advantages over a model calculation. This is to one part that there are no changes in circulation considered (the extend of which are not well known) and that EESC is based to a very large degree on observations. Not only the tropospheric time series are observation based but also the chemical loss (fractional release factors). With respect to EESC projections not including changes in circulation makes it a valid tool to assess the rate of recovery of the parameter which is most easily influenced by anthropogenic activity and in end political action, namely the halogen loading. Of course, this changes is for "all other things being equal". The suggested comparison with model calculation is far from trivial, as in these model calculation ESC (equivalent stratospheric chlorine, e.g. the sum of $Cl_y + 60$ or 65 times $Br_y$ from the model) is dependent on many more factors, especially the expected changes in stratospheric transport. These changes can only be disentangled in model calculations without long term changes in circulation but with changes in trace gases, e.g. calculations with fixed dynamics. We have performed such a comparision for a model with changing dynamics (EMAC) and a model with fixed dynamics (TOMCAT). The comparision is included in a new section 4.3. The change in dynamics in the EMAC model results in large differences between EESC and ESC when ESC is evaluated on a given mean age level, but reasonable agreement when ESC is evaluated on a fixed pressure level. The comparison with the TOMCAT model and the old formulation of EESC and our new formulation shows that a much better agreement is found using the new formulation. This is discussed in the new section 4.3. and the conclusions section, as explained in the answer to the major question #3.*

**Reviewer 2**

This paper describes a reformulation of the EESC metric that attempts to account for the difference in stratospheric mass transport through photochemical loss regions vs. transport outside of the loss regions. The result is a relatively older mean age used to calculate EESC and thus a delay in future mid-latitude lower stratospheric EESC values to 1980 levels of roughly one decade.

Unfortunately, the paper has a number of fundamental flaws that make it unsuitable for publication. The major issues include (1) the absence of path height dependence in the fractional release and modified age distributions, (2) the assumption of constant stratospheric dynamics and photochemistry over nearly a century for the results to be valid, (3) the relatively minor role of EESC in future mid-latitude ozone depletion compared to N2O and (4) the absence of many relevant references and discussion of previous work.

Main comments:

1. The authors mention in Section 2 that chemical loss is not uniform throughout the stratosphere and argue that on average the longer a fluid element remains in the stratosphere the larger the integrated chemical loss will be. This is not the case. The highly nonlinear loss dependence on altitude and latitude make the path an air parcel has taken much more important than the time an air parcel has resided in the stratosphere. Hall (2000) described the concept of maximum path height and the relationship between the path height and age distribution. Hall showed quite clearly that the mass fraction of an air parcel that has passed above some height, such as the height above which photochemical loss is rapid, is determined by the transport due to mass continuity and not the circulation, including mixing, that determines age distributions. Thus, the ge distribution is only weakly linked to stratospheric photochemical destruction of any trace gas. This means that it is the circulation due to mass continuity that is most relevant in the estimation of EESC.

The circulation due to mass continuity is essentially the residual circulation and it has been shown, such as by Birner and Bonisch (2011) that the transit time due to the residual circulation is at most 3.5 years in the polar regions where the air parcels reached a minimum pressure of less than 0.1 hPa, well into the mesosphere, before descent into the polar vortices. This also implies that transit times throughout the mesosphere are actually less than 3.5 years. Mixing of air horizontally acts to increase the age of the stratosphere nearly everywhere (e.g. Garny et al., 2014) but not necessarily the maximum path height of a parcel and thus it's photochemical loss. The mixing is what drives the old tail in the age distribution and this mixing is not directly correlated with changes in path height. The more appropriate time scale for photochemical loss is the mean arrival time at the location(s) where each trace gas is rapidly destroyed. Once an air parcel passes through a region of rapid photochemical loss for a particular tracer the first time then all of that tracer is destroyed (converted into chlorine and bromine in this case) and it doesn't matter what happens to the air parcel from then on as far as the fractional release. Subsequent aging has no further effect on the release of chlorine or bromine. The mean arrival times at the region of rapid photochemical destruction for each trace gas will be dependent on the stratospheric transport each year and will be variable.

*This comment is to a large part in line with the comments by reviewer #1, except that here the region of rapid photochemical loss is emphasized, whereas reviewer #1 uses the maximum path height (MPH) as a proxy for this. As explained above, we do not want to state that transit time is the only factor, but we think that there are good arguments that transit time is an important factor and that **on average** the time in the rapid photochemical loss region increases with transit time. Similar as explained above we included a more detailed discussion of this in the text:*

*The fractional chemical loss can be expressed in a very generalized way as (1-f(t^',p)), where f(t^',p) is a fractional release function, which is specific for each trace gas and will depend on the time the air parcel has spent in the stratosphere t' and on the path p it has taken during the transport, especially also of the maximum path height (MPH) (Hall, 2000). is While the path or the MPH are not known, it has been shown that "molecules arriving at X with long arrival times will, on average, have spent more time exposed to chemical loss and will have sampled atmospheric regions where photochemical loss is greater" (Plumb et al., 1999). We therefore follow the approach that the fractional loss can on average be described as function of the transit time only. That chemical loss and transit time are on the average related to each other is also reflected in the tight observed correlations between mean age and tracer mixing ratios (e.g.Volk et al., 1997;Engel et al., 2002). While there will be fluid elements with very different paths and different chemical loss which have the same transit time the loss can on average be sufficiently well described as a function of the transit time. We therefore treat f(t^',p) as f(t^' ) only. This is also in line with the findings of Schoeberl et al. (2000), who showed that using an "average path approximation" with a "single-path photochemistry" and thus with a unique relationship between loss and transit time, global tracer-tracer correlations can be explained. This concept that loss can be described only as a function of transit time without considering the different transit*

*pathways was also adopted by Schoeberl et al. (2005) in the derivation of age spectra. We use a mean age of 3 years for mid latitudes and of 5.5. years for high latitudes. Indeed, the path distribution for an air parcel with mean age of 3 years in the tropics, in mid latitudes and in polar regions is expected to show more variability than for air parcels investigated under similar conditions (e.g.latitude regions). As this analysis is restricted to one latitude band for one mean age level, we therefore*
*approximate loss as a function of transit time only.*

2. The assumption of constant stratospheric dynamics and photochemistry from the late 20th century to the late 21st century is certainly not a good one. What is the sensitivity of the results to the predicted changes in the stratospheric circulation?

*If the circulation accelerates, as projected by all climate models, we expect an increase in fractional release on a given mean age surface and also an upward movement of mean age surfaces in the stratosphere. The exact extend is strongly dependent on changes in residual circulation and in mixing and this will certainly affect fractional release and thus inorganic halogen loading. We have included a comparison between our new EESC calculation and ESC from a chemistry climate model (EMAC) In the new section 4.3. Based on this we conclude that EESC is a reasonable proxy for ESC on a constant pressure level, in*
*agreement with findings by Shepherd et al.*

*We have added the following in the introduction:*

*Note that EESC is only a valid proxy for anthropogenic ozone depletion if all other parameters, especially atmospheric*
*transport, are unchanged. A projection of a return of EESC to some specific level therefore does not imply that ozone will return to the same levels. EESC should thus be regarded as proxy for the impact of halogenated source gases on the ozone layer due to both anthropogenic and natural emissions. The recovery of the ozone layer is affected by other parameters in addition, especially changes in transport. EESC is therefore not a proxy for ozone recovery, but a proxy for the impact due to one single parameter, the halogen loading. In section 4.3. we discuss the validity of EESC as a proxy for inorganic halogen*
*loading under the influence of changing stratospheric dynamics.*

*We have further added more on this issue in the new section 4.3., especially the section on*

**Comparison to model calculations with varying dynamics**

*Under changing stratospheric dynamics (e.g.Butchart, 2014), it is expected that fractional release factors at a given mean age level will change (Douglass et al., 2008;Li et al., 2012;Ostermöller et al., 2017). Therefore, the inorganic halogen loading as a function of mean age would be expected to change even if all source gases remained constant in time. Under such conditions, EESC is not expected to follow ESC on a given mean age level. To estimate the validity of EESC as a proxy for inorganic halogen loading of the stratosphere, we have compared our new formulation to a free running chemistry climate model*
*simulation. We used data from the EMAC model simulation RC2-base-04 from the ESCiMo project for this (Jöckel et al., 2016). This simulation covers the 1950–2100 time frame with simulated sea surface temperatures and sea ice contents. As described above, we again calculated fractional release factors from the model in order to have results which are internally consistent. Northern Hemisphere fractional release factors for the year 2000 are in good agreement with observation based fractional release factors (Newman et al., 2007;Laube et al., 2013) and therefore we used northern hemispheric data for this*
*comparison. In addition to comparing ESC on a fixed mean age level we also compared ESC on a fixed pressure level to our new formulation of EESC. A similar comparison has been presented in Shepherd et al. (2014), who compared model ESC on a fixed pressure level to EESC on fixed mean age level (using the formulation of Newman et al. (2007)), showing good agreement. Figure 9 compares the time evolution of EESC for 3 years of mean age with model ESC at the 60 hPa level (corresponding to 3 years of mean age in the year 2000) and model ESC at 3 years of mean age. The year of 2000 and the*
*corresponding level of 60 hPa was chosen, as we also evaluated fractional release factors in the year 2000 of the model run. As expected, ESC at 3 years of mean age deviates systematically from EESC, especially in the future when fractional release evaluated on a mean age surface changes significantly in the model. The agreement with ESC on a fixed pressure level is*

*however much better. In this comparison EESC at 3 years of mean age would slightly overestimate ESC on a pressure level in the future and significantly underestimate ESC on a mean age level. The exact magnitude of changes in stratospheric dynamics is highly uncertain and also we expect ozone to follow a pressure surface rather than a mean age surface in the future. We therefore conclude that EESC is a reasonable proxy for the effect of halogen loading on stratospheric ozone, given the overall high uncertainties associated to the future evolution of stratospheric dynamics.*

*In the discussion section we have added the following:*

*We have shown that he long-term evolution of equivalent stratospheric chlorine (ESC, i.e. inorganic chlorine and bromine, the latter weighted in a similar way as in EESC to reflect the higher efficiency of bromine to ozone depletion) in the model deviates substantially from our calculation of EESC in a long-term model calculation with varying dynamics. However, we have also shown that the new formulation of EESC is a reasonable proxy for the evolution of inorganic halogen loading on a given pressure level. We therefore conclude that EESC is a reasonable proxy for future halogen impact on ozone.*
*…*
*As current climate models consistently predict an acceleration in the Brewer-Dobson circulation (Butchart, 2014), this will have an impact on the temporal evolution of inorganic halogen loading of the stratosphere. These expected changes in the Brewer-Dobson circulation would result in an earlier recovery of ozone at mid- and high latitudes (Eyring et al., 2010). These changes are not included in the concept of EESC. However, we have shown that EESC is a reasonable proxy for ESC when ESC is evaluated at constant pressure level.*

3. $N_2O$ is only mentioned briefly in the conclusions but mid-latitude ozone depletion in the late 21st century will be due primarily to $N_2O$ concentrations (Ravishankara et al.,2009, Portmann et al., 2012, Butler et al., 2016). The variability and uncertainty in the N2O concentrations will be much more of a factor in the return of mid-latitude ozone to 1980 levels than the small variability in the decline of EESC.

*The reviewer claims that ozone depletion in the 21st century will be primarily due to N2O concentrations. This is not correct. The paper by Ravishankara states that in the 21st century the emissions of $N_2O$ are more important than the emissions of any individual CFCs. The ozone depletion due to chlorine concentrations is nevertheless still the dominant anthropogenic influence on ozone. In addition, $N_2O$ cannot just be included in the same way as chlorine and bromine, as its efficiency for ozone depletion changes with changing halogen loading (Daniel et al., 2010). This would add another level of complexity, which is not the main focus of this study. In order to make this clear, we added the following in the introduction section of the paper.*

*While in principle, $N_2O$ could be included in EESC, as the $NO_x$ released in the stratosphere also leads to ozone depletion (Ravishankara et al., 2009), this is complicated, as the efficiency with which $N_2O$ leads to ozone depletion changes with halogen loading (Daniel et al., 2010;Ravishankara et al., 2009). We therefore do not include $N_2O$ in EESC.*

4. All of the above references are relevant to this study and none of them were included. The study of Waugh et al. (2007) is also highly relevant since they explore and discuss the age vs. path sensitivity of inorganic chlorine in the stratosphere. A further point, what is special about recovery to 1980 levels? As was discussed in Newman et al. (2007) the year chosen to be the initial year causes large variability in the recovery time due to the steep slope of EESC around 1980 and the gradual slope in the late 21st century.

*We thank the reviewer to pointing us to further studies and apologize for not including them. We have included more references*

*to further work on links between dynamical changes and chemical changes and also on the factors influencing chemical loss.*

*In the introduction we have added:*

*In models which include both chemistry and transport of the stratosphere, the amount of inorganic halogen can be directly*

*calculated as $Cl_y$ and $Br_y$. On average, the inorganic halogen content is higher for air parcels with higher mean age (Newman*

*et al., 2007;Engel et al., 1997), but the relation between $Cl_y$ and mean age may differ from model to model depending on the representation of transport and chemistry in the model (Waugh et al., 2007). The relation between mean age and $Cl_y$ depends on the interaction between transport and chemistry and is a function of both the time spent in the stratosphere and the transport pathways (Hall, 2000;Schoeberl et al., 2000;Schoeberl et al., 2005;Waugh et al., 2007).*

*Further in the section 2 which describes the influence of transport and tropospheric trends on chemical active species, we have added a detailed discussion of the dependency, including the relevant references:*

*The fractional chemical loss can be expressed in a very generalized way as (1-f(t^',p)), where f(t^',p) is a fractional release function, which is specific for each trace gas and will depend on the time the air parcel has spent in the stratosphere t' and on the path p it has taken during the transport, especially also of the maximum path height (MPH) (Hall, 2000). is While the*

*path or the MPH are not known, it has been shown that "molecules arriving at X with long arrival times will, on average, have spent more time exposed to chemical loss and will have sampled atmospheric regions where photochemical loss is greater" (Plumb et al., 1999). We therefore follow the approach that the fractional loss can on average be described as function of the transit time only. That chemical loss and transit time are on the average related to each other is also reflected in the tight observed correlations between mean age and tracer mixing ratios (e.g.Volk et al., 1997;Engel et al., 2002). While*

*there will be fluid elements with very different paths and different chemical loss which have the same transit time the loss can on average be sufficiently well described as a function of the transit time. We therefore treat f(t^',p) as f(t^' ) only. This is also in line with the findings of Schoeberl et al. (2000), who showed that using an "average path approximation" with a "single-path photochemistry" and thus with a unique relationship between loss and transit time, global tracer-tracer correlations can be explained. This concept that loss can be described only as a function of transit time without considering the different transit*

*pathways was also adopted by Schoeberl et al. (2005) in the derivation of age spectra. We use a mean age of 3 years for mid latitudes and of 5.5. years for high latitudes. Indeed, the path distribution for an air parcel with mean age of 3 years in the tropics, in mid latitudes and in polar regions is expected to show more variability than for air parcels investigated under similar conditions (e.g.latitude regions). As this analysis is restricted to one latitude band for one mean age level, we therefore approximate loss as a function of transit time only.*

**Reviewer #3**

The paper studies the combined effect of chemical loss and transport for the formulation of a simplified index describing the ozone depleting capacity due to halogens in the atmosphere. Usually, for that purpose the quantity EESC is used (for example in the WMO ozone assessments). EESC experienced several changes in its definition already in the past, when the correction for transport times was refined including its impact on release factors and the impact of transport time distribution when correcting for a non-linear trend has been included. Following Newman et al. (2007), the authors here present a further refinement of this concept. Generally, I find this paper a very nice exercise to better understand how chemical loss and transport work together, here applied to describe the halogen loading in the atmosphere. The value of the paper is to remove inconsistencies in the formulation of the traditional EESC concept, but still use the same simplified approach. The paper is clearly written, references are given as necessary, the assumptions are discussed in detail and possible consequences for the ozone recovery description are discussed.

My main obejection is the title of the paper which is in my opinion totally misleading. The paper is about a refinement of the EESC concept. In this concept, parameters can be derived like the time of recovery. This parameters has a descriptional character, and can be used to compare different scenarios of Halogen loading, for example. Modifiying the formulation of the concept will change the value of a parameter, but not the scenario, and therefore also not its "real" Halogen loading. In addition, as the authors state by themselves, the concept does not include deviations of the implicit assumed stationarity of the dynamics which is not even true for the past.

*We completely agree; the title was really misleading. We have renamed the paper in order to emphasize that the main goal is to derive a new formulation of EESC and that we then discuss some implications of this formulation. The title has been changed to "A refined method for calculating Equivalent Effective Stratospheric Chlorine".*

My second concern is somewhat related: as the EESC concept describes only one of the main driving processes for additional ozone depletion I would ask the authors therefore to put their results in the context of model studies where the effect of the accelerating Brewer Dobson circulation has been analysed.

*We have now included more discussion on the link between EESC and changes in the Brewer Dobson circulation. In the introduction:*

*We have further added more on this issue in the discussion (see also answers to reviewer 2 and 3):*

*Note that EESC is only a valid proxy for anthropogenic ozone depletion if all other parameters, especially atmospheric transport, are unchanged. A projection of a return of EESC to some specific level therefore does not imply that ozone will return to the same levels. EESC should thus be regarded as proxy for the impact of halogenated source gases on the ozone layer due to both anthropogenic and natural emissions. The recovery of the ozone layer is affected by other parameters in addition, especially changes in transport. EESC is therefore not a proxy for ozone recovery, but a proxy for the impact due to one single parameter, the halogen loading. In section 4.3. we discuss the validity of EESC as a proxy for inorganic halogen loading under the influence of changing stratospheric dynamics.*

*We have further added more on this issue in the new section 4.3., especially the section on*

**Comparison to model calculations with varying dynamics**

Under changing stratospheric dynamics (e.g.Butchart, 2014), it is expected that fractional release factors at a given mean age level will change (Douglass et al., 2008;Li et al., 2012;Ostermöller et al., 2017). Therefore, the inorganic halogen loading as a function of mean age would be expected to change even if all source gases remained constant in time. Under such conditions, EESC is not expected to follow ESC on a given mean age level. To estimate the validity of EESC as a proxy for inorganic halogen loading of the stratosphere, we have compared our new formulation to a free running chemistry climate model simulation. We used data from the EMAC model simulation RC2-base-04 from the ESCiMo project for this (Jöckel et al., 2016). This simulation covers the 1950–2100 time frame with simulated sea surface temperatures and sea ice contents. As described above, we again calculated fractional release factors from the model in order to have results which are internally consistent. Northern Hemisphere fractional release factors for the year 2000 are in good agreement with observation based fractional release factors (Newman et al., 2007;Laube et al., 2013) and therefore we used northern hemispheric data for this comparison. In addition to comparing ESC on a fixed mean age level we also compared ESC on a fixed pressure level to our new formulation of EESC. A similar comparison has been presented in Shepherd et al. (2014), who compared model ESC on a fixed pressure level to EESC on fixed mean age level (using the formulation of Newman et al. (2007)), showing good agreement. Figure 9 compares the time evolution of EESC for 3 years of mean age with model ESC at the 60 hPa level (corresponding to 3 years of mean age in the year 2000) and model ESC at 3 years of mean age. The year of 2000 and the corresponding level of 60 hPa was chosen, as we also evaluated fractional release factors in the year 2000 of the model run. As expected, ESC at 3 years of mean age deviates systematically from EESC, especially in the future when fractional release evaluated on a mean age surface changes significantly in the model. The agreement with ESC on a fixed pressure level is however much better. In this comparison EESC at 3 years of mean age would slightly overestimate ESC on a pressure level in the future and significantly underestimate ESC on a mean age level. The exact magnitude of changes in stratospheric dynamics is highly uncertain and also we expect ozone to follow a pressure surface rather than a mean age surface in the future. We therefore conclude that EESC is a reasonable proxy for the effect of halogen loading on stratospheric ozone, given the overall high uncertainties associated to the future evolution of stratospheric dynamics.

*In the discussion section we have added the following:*

*We have shown that he long-term evolution of equivalent stratospheric chlorine (ESC, i.e. inorganic chlorine and bromine, the latter weighted in a similar way as in EESC to reflect the higher efficiency of bromine to ozone depletion) in the model deviates substantially from our calculation of EESC in a long-term model calculation with varying dynamics. However, we*

*have also shown that the new formulation of EESC is a reasonable proxy for the evolution of inorganic halogen loading on a given pressure level. We therefore conclude that EESC is a reasonable proxy for future halogen impact on ozone.*
*…*
*As current climate models consistently predict an acceleration in the Brewer-Dobson circulation (Butchart, 2014), this will have an impact on the temporal evolution of inorganic halogen loading of the stratosphere. These expected changes in the*

*Brewer-Dobson circulation would result in an earlier recovery of ozone at mid- and high latitudes (Eyring et al., 2010). These changes are not included in the concept of EESC. However, we have shown that EESC is a reasonable proxy for ESC when ESC is evaluated at constant pressure level.*

The derivation of the concept is somewhat lengthy in my opinion and can be combined, for example for equs. (11)-(14). Stationarity means that at the end that the combined history of an air parcel (including its mean photochemical dose) is only a function of the position in the atmosphere. So equation (19) is not a surprise. Much more subtle is the transition from equ. (19) to (20). This is valid only under the specific condition that the distribution G# is determined by its first moment only which may be not generally true. Here I would ask the authors to discuss the assumptions in more detail.

*As none of the other reviewers has commented on this derivation being too long, we would rather keep it in its present form. The transition from (19) to (20) does indeed imply that all locations where mean release time has the same value, the release time distribution is the same. This is very closely linked to the discussion of path dependency (see answers to reviwer #1 and*

*#2 and general introduction to our answers). In order to make this underlying assumption we have included the following statement in the explanation of the transition from (19) to (20):*

*This implies that at all locations r with the same mean release time, the release time distribution is the same. This assumption may not be valid everywhere, but as a mean age of 3 years is used for mid latitudes and of 5.5 years for high latitudes, we use this assumption only for air parcels under similar meteorological conditions (latitude bands).*

Finally, to be more than just an interesting exercise, the paper would strongly improve if the authors could show that using their new formulation would yield a more concise ozone trend analyses, at least in one example

*It is not the intention of this paper to repeat studies using EESC as a proxy for halogen-induced ozone loss. In any case, as the timing of e.g. the maximum is rather similar in both formulations, we do not believe that the results would be very different.*
*In order to emphasize this we have included the following in the conclusions:*

*As the suggested reformulation of EESC does not affect the principal behavior of the temporal evolution of EESC, we do not expect this reformulation to lead to substantial changes, which could impact the changes of studies using EESC except for those which have used EESC to project EESC recovery.*

**Minor points**

p2l9: transport within: *changed to into and within*

p4l17: you mean averaged over the seasons == annual mean: *seasonal mean changed to annual mean*

p4l18: eliminate "it is expected": *deleted*

p4l20: the typical path from the tropical tropopause through the stratosphere back to the troposphere at higher latitudes will
not yield this shorter lifetime at the end of the path. You mean strictly in the stratosphere.

*Yes, and actually this should also be restricted to the tropical latitudes. We added: "and topical latitudes in the stratosphere"*

p4l21: f will be a function of r, too. This does not harm the derivation.
*This is the linked to the main point raised by reviewer 1 and 2. We added a discussion on the additional dependence on the location, or more specifically the transport pathway. Concerning the dependence on the location, we have added the following:*

*The fractional chemical loss can be expressed in a very generalized way as $(1-f(t^{'},p))$, where $f(t^{'},p)$ is a fractional release*
*function, which is specific for each trace gas and will depend on the time the air parcel has spent in the stratosphere t' and on the path p it has taken during the transport, especially also of the maximum path height (MPH) (Hall, 2000). is While the path or the MPH are not known, it has been shown that "molecules arriving at X with long arrival times will, on average, have spent more time exposed to chemical loss and will have sampled atmospheric regions where photochemical loss is greater" (Plumb et al., 1999). We therefore follow the approach that the fractional loss can on average be described as*
*function of the transit time only. That chemical loss and transit time are on the average related to each other is also reflected in the tight observed correlations between mean age and tracer mixing ratios (e.g.Volk et al., 1997;Engel et al., 2002). While there will be fluid elements with very different paths and different chemical loss which have the same transit time. on the average the loss can be sufficiently well described by the transit time. We therefore treat $f(t^{'},p)$ as $f(t^{'})$ only. This is also in line with the findings of Schoeberl et al. (2000), who showed that using an "average path approximation" with a "single-path*
*photochemistry" and thus with a unique relationship between loss and transit time, global tracer-tracer correlations can be explained. This concept that loss can be described only as a function of transit time without considering the different transit pathways was also adopted by Schoeberl et al. (2005) in the derivation of age spectra. We use a mean age of 3 years for mid latitudes and of 5.5. years for high latitudes. Indeed, the path distribution for an air parcel with mean age of 3 years in the tropics, in mid latitudes and in polar regions is expected to show more variability than for air parcels investigated under*

*similar conditions (e.g.latitude regions). As this analysis is restricted to one latitude band for one mean age level, we therefore approximate loss as a function of transit time only.*

p5l31: what are the three? I see the trend and the chemical loss only.

*The three function are the age spectrum, the loss function and the temporal trend. We have changed the order slightly to have the three functions immediately after each other:*
*... must be considered, which are all functions of the transit time. We will denote transit time, i.e. the time a fluid element has spent in the stratosphere as t', while the time itself will be denoted as t. First, the transit time distribution, i.e. how long it has taken for the individual fluid elements of this air parcel to travel from their entry point to the stratosphere to the location r in the stratosphere. Second, the temporal trend of the mixing ratios at the entry point has to be considered and, third, chemical loss during this transport.*

p6l2: the exponential chemical loss term is only used here and can be left out.
*Yes, we have left out the exponential term and used the fractional lost directly.*

*… and the chemical loss term, which can be be described by the factor $\left(1 - f(t')\right)$, where $f(t')$ describes the fraction which has been lost.*

$$\chi_{strat}(r,t) = \int_0^\infty \chi_0(t - t') \cdot \left(1 - f(t')\right) \cdot G(r,t')dt' \qquad (6)$$

*As $\left(1 - f(t')\right)$ is the remaining fraction of the organic …..*

p11l9 remove "classical": *removed*

Typos:
parameterization/parameterization should be typed in one version only

*all changed to parameterization*

p3l9: the first moment has a lower value: *changed*

**Reviewer #4**

This manuscript has built nicely on Ostermoller et al. (2017). The concept developed there is used to derive a relationship between previously calculated fractional release (FRF) values that assumed an age spectrum representative of an inert tracer to FRF values that are independent of tropospheric source gas trends. More importantly, this work quantifies the importance of using an age spectrum that accounts for chemical loss when calculating equivalent effective stratospheric chlorine (EESC). This improved approach effectively leads to older air in the EESC calculation, particularly for the midlatitude stratosphere. This, in turn, implies lower EESC values in 1980; this 1980 level has been important because it has typically been taken as a value of significance in the return of stratospheric chlorine/bromine to natural levels. The proposed EESC revision (i.e., older air) also leads to higher EESC values for any given time when source gases are declining. These changes combine to lead to a substantial delay in the time when mid-latitude EESC is projected to return to 1980 levels. As expected, the effects are smaller for polar EESC, since the difference in the average age for the dissociated ODSs and an inert tracer are much reduced.

I have a few general comments here, and some more specific ones below. Assuming these comments can be dealt with sufficiently, I find this manuscript to be valuable and I believe that it should offer an important improvement on work that came before it. It would be useful to describe whether EESC from the new formalism is distinct enough from EESC using the old one so that past work that used EESC should have identified a shortcoming in the previous approach. Looking at Figure 3,

I would be particularly interested in previous work that compared measurements or model calculations over a time range that spanned both before and after the EESC peak in the late 1990's, since the differences should be most apparent over such a period. If the two approaches are not distinct enough to be apparent in previous work, this would be worth stating here, so the reader knows the main impact is on the "recovery" date, and that it doesn't affect the validity of previous results.

*As the timing of e.g. the maximum is quite similar in both formulations, we do not believe that the results would be very different. In order to emphasize this we have included the following in the conclusions:*

*As the suggested reformulation of EESC does not affect the principal behavior of the temporal evolution of EESC, we do not expect this reformulation to lead to substantial changes, which could impact the changes of studies using EESC except for*
*those which have used EESC to project EESC recovery.*

The only other comment I particularly want to highlight here relates to the sensitivity study of the width of the transport distribution function. Please see my comment below for page 14, lines 2-3. I would find this most useful if you explored the impact of a change in width of the age distribution relevant to an inert tracer, with that impact propagating to the halocarbons
depending on their chemical loss; however, unless I am mistaken, it doesn't seem like this is what is done.

*See our reply to this specifically below (answer to comment on p 14, l.2-3.)*

Specific comments:
Page 1, Line 1 At some point, relatively early in the manuscript, you should make clear what you are not implying by this title, otherwise it could be considered misleading. As currently written, it could be taken to suggest that there has been more ODS emission than expected or that dynamics may change in an unexpected way to alter halogen loading in the future. An alternative that may be preferable would be to change to a title more focused on the delay in EESC recovery.

*As noted by all reviewers (and we completely agree), the title was misleading. The new title "A refined method for calculating Equivalent Effective Stratospheric Chlorine" does not put the focus on an estimated recovery date anymore.*

1, 16 1980 is not the year of stratospheric ozone depletion onset, but it is often used as a benchmark to measure significant progress towards recovery

*Rephrased to "used a benchmark to which EESC must return in order to reach significant progress towards halogen and ozone recovery."*

3,16 I suggest clarifying what 'this purpose' refers to at the end of this sentence

*The sentence has been changed to clarify this: "The age spectrum for in inert tracer is not well suited to describe the propagation of a tracer with chemical loss into the stratosphere."*

5,1-3 It is not clear to me that this sensitivity study addresses the entire phase space of possibilities in your assumed relationship
between age and loss. Additional justification is needed to show that the simple relationship you are basing your calculations on are sufficiently appropriate.

*The Figure showing the different transit time distributions is purely for illustrative purposes and makes no claim that these are the real distributions. As explained in the manuscript (section 4), we have varied several parameters in order to test the*
*influence on EESC calculated using our new formulation and found that the impact on EESC and on projected recovery dates*

*is not very large. The best way to show that the relationship used here is sufficiently appropriate would be to compare EESC with **observations** of inorganic chlorine and bromine. (Plumb et al., 1999) showed that it gives very much improved representation of organic species, which is a good indication that the inorganic fraction should also be better described. Such observations of inorganic chlorine and bromine are not available, which is why we have chosen to test our new formulation in a full model calculation. A new section on this comparison has been added (section 4.3., especially the first part on the comparision with a fixed dynamics model calculation), which shows that the new formulation yields much improved agreement with model calculations, although certain discrepancies remain. We have also added in the conclusions to the paper that the relationship between loss/release and transport should be explored in more realistic models.*

*Such calculations are only available based on a rather old 2D model (Plumb et al., 1999) and should be repeated with state-of-the-art models.*

6, 2 It is not clear to me that having the loss described as an exponential term with the lifetime depending on transit time is helpful in the formulation. It is really of an arbitrary mathematical form since the lifetime (denominator) varies with the location endpoint. It would seem more straightforward to skip straight to the factor $(1-f(t'))$, but I leave this decision to the authors.

*Yes, the exponential function has been deleted and we now go straight to the factor $(1-f(t'))$*

10, 19 Somewhere you should discuss the impact of using the Plumb age estimates from an old 2-D model given the advancements in our ability to calculate circulation metrics over the last 20 years and the general superiority of 3-D models at making these calculations today

*The reviewer is correct in pointing this out. We had included in the conclusions, that the parameterizations should be revisited with state of the art models. In order to make this clearer, we have added a statement on this in the new subsection Sensitivity to the mean release time derived from mean age and stratospheric lifetime:*

*Despite this rather low sensitivity, it should be noted that the parameterization is derived from a 2D model. The relationship between mean age, age spectrum and chemical loss should be explored in state-of-the-art 3 D models, which have a better representation of stratospheric transport processes.*

*In the place highlighted by the reviewer, we have added the following:*
*Plumb et al. (1999) used a 2D model for their study. Despite this, the stratospheric lifetimes derived from the model are in overall good agreement to more recent model studies (Chipperfield et al., 2013). The sensitivity of the parameterization between mean release time and mean age to the stratospheric lifetimes is further discussed in section 4.*

14, 2-3 I am having trouble understanding exactly what is being done here. Are the factors changed for both G and G#subN? It would not be possible to have gamma be 0 for G#subN and be 0.7 for G, would it? But if both gamma factors are 0, it would seem that this approach would collapse to the old result since the mean release age would be the same as the mean age of an inert species. And if that is the case, I would have expected a larger impact on return times (i.e., they should be close to the VD (2014) values). Perhaps it would help if you had a figure (like Figure 1) showing what the G curves look like as gamma goes to 0 and for it equal to 2. It looks like you may be using the values from Table 1; however, this doesn't seem appropriate if you want to examine the impact of a changing shape in the overall transport distribution function. In fact, I'm unclear physically what is going on here, so clarification would be very helpful.

*Thank you for this suggestion, which we have considered. As Figure 1 is however only for illustrative purposes and does not claim to be represent the real loss time and arrival time distributions, we think it would give too much weights to this Figure if we included it for other parameterizations. Therefore, we would rather not include additional similar Figures.*
*Concerning the comment on the range of lamda factors (we think this is what the reviewer is commenting on, not gamma values, as we varied lambda between 0 and 2) included in the sensitivity study, the reviewer is correct, that in case of a gamma*

*value of 0 for the age spectrum, the mean release time would be the same as the mean age. We did not apply this sensitivity study to the derivation of mean arrival time from mean age. This sensitivity is investigated by varying the assumed stratospheric lifetimes. The sensitivity test described here is only applied to $G_N^\#$. In the new method for the calculation of EESC there is actually no G anymore, just the $G_N^\#$ distributions for the various tracers. So it is only these distributions which are varied.*

*Indeed, a gamma factor of 0 for $G_N^\#$ does not make a lot of sense, since this would mean that there is no mixing at all and this would make the whole concept of an age spectrum obsolete. But it could be regarded as a limiting case: the age spectrum cannot possible be any more narrow than this, as a gamma factor of 0 is equivalent to a pure propagation without mixing. On the other hand, a gamma value of 2 is equivalent to a very wide spectrum. Therefore, the tested values do indeed represent extreme cases and the purpose was to show that the sensitivity to the exact shape of $G_N^\#$ is rather low, but that the main*

*sensitivity is to the first moment of the distribution, which is the mean release time. We have broken down this subsection into two subsections, one dealing with the sensitivity to the shape of the release time distribution and one dealing with the sensitivity due to the derivation of the mean release time. These section are now called "Sensitivity to the shape of the new release time distribution $G_N^\#$ " and "Sensitivity to the mean release time derived from mean age and stratospheric lifetime". To specify that we only varied the shape of $G_N^\#$ , we have added this in the title of the subsection "Sensitivity to the **shape of the new release***

***time distribution $G_N^\#$****" and also in the text::*

*... values of 0 and 2 years in the calculation of $G_N^\#$, while retaining the first moment, i.e. $\Gamma^\#$. The ..*

14, 25-29 Please describe how the destruction vs. age relationship is determined for these perturbations

*The perturbations are simply introduced by changing the stratospheric lifetime in the formula given by Plumb et al., for the calculation of mean arrival time from mean age and stratospheric lifetime. We have added a further description to make this clear:*

*We tested the sensitivity of our calculation to that by systematically increasing all lifetimes by 20% or decreasing them by 20% (see Table 5) in the parameterization given by (Plumb et al., 1999). This results in different mean arrival time $\Gamma^*$ and mean release time $\Gamma^\#$.:*

15, 15 Perhaps broaden this statement, if you think it is accurate to something like "This approach more accurately represents the amount of Cly and Bry in the stratosphere from tropospheric source gas concentrations, and should be adopted to estimate. . ." This would seem to be more consistent with the title, but do please refer to my earlier concerns of such a broadening.

*As noted before, we have changed the title of the paper. Nevertheless we agree with the reviewer and have added the suggested*

*statement.:*

*... calculated by the method of Ostermöller et al. (2017). This approach more accurately represents the amount of $Cl_y$ and $Br_y$ in the stratosphere from tropospheric source gas concentrations and fractional release factors. We suggest it should be adopted to calculate EESC and to estimate ...*

Minor comments:
1, 10 Do you mean 'adopted' here? While it is adapted through your work, that doesn't seem to be the intent here

*Changed to adopted*

1, 18 Change 'assumed' to 'estimated'

*Changed to estimated*

1, 27 Change to 'winter and springtime': *changed*

2,1 Replace 'effectiveness' with 'extent' or something similar; otherwise it could sound like the destruction per Cl molecule is what you are referring to here

*Changed to extent*

2,2 Reference EESC: *done, referenced Daniel and Velders 2011 and Newman et al., 2007.*

9, 1 Change 'was' to 'way': *changed*

9, 3 1-f(t) doesn't seem to be appropriately named as the 'loss term'; I understand why you called it the 'chemical loss term' back in eq. (6), but now in isolation is seems confusing that it is a 'loss term' that is really equal to one minus the fractional loss

*We are not sure if we understand this comment correctly. We refer to f(t') as the **release** (not loss) term and to 1-f(t') as the loss term. We believe that this is correct.*

12, 5 Change 'compares' to 'compare': *changed*

12, 27 Change 'on the new' to 'of the new' at the end of the line: *changed*

13, 12 Add space in 'STRATcampaign': *done*

13, 23-24 At first, this sentence seemed to suggest that you were doing another calculation from the Ostemoller results, but in fact, I believe you are summarizing the 2060 vs 2058 discussed at the top of this page. This could be clarified.

*Upon rereading, we agree. We deleted this sentence as it only repeats what is said before and adds more confusion than clarification.*

14, 18 Change to 'independent of';: *changed*

15, 5 I don't understand the use of the word 'respective' here: *changed to organic or inorganic*

15, 8 This seems to not be an appropriate use of 'Therefore'. From what is stated here, the following sentence doesn't seem to logically follow the previous statement(s).

*We have rephrased as follows for more clarity: Fractional release factors which are independent of the tropospheric trends (Ostermöller et al., 2017) must be used to correctly*

15, 16/17 I suggest changing to present tense: *done*

15, 20 Change to '. . .perturbed values of stratospheric chlorine and bromine. . .' changed to "… the mid-latitude lower stratosphere to unperturbed values of chlorine and bromine" …

15, 21-23 You should probably also point out that CO2 is expected to accelerate column
ozone recovery across much of the globe (see, e.g., Butler et al., 2016 and many
others)

*The reviewer is referring to the expected acceleration of the Brewer-Dobson circulation due to climate change. We have given this issue much more weight in several parts of the manuscript. We have further added a statement on ozone recovery here, referencing a paper on ozone recovery from many models (Eyring et al., 2010):*

*These expected changes in the Brewer-Dobson circulation would result in an earlier recovery of ozone at mid- and high latitudes (Eyring et al., 2010). In addition to this, increases in the concentrations of $N_2O$ and short-lived chlorine containing halocarbons may further influence the recovery of the ozone layer, possibly leading to a later recovery (Hossaini et al., 2015b;Hossaini et al., 2015a;Chipperfield, 2009).*

15, 33 It could be useful to say what G# is here, so people who didn't read the main
text will know what it is.
"of the loss weighted transit distribution" has been added as explanation

16, 10 Make 'distributions' singular. *done*

16, 17 Change 'calculation' to plural. *done*

Table 1 For CFC-113 and CH3Br add another significant figure '0' to the end of the time-independent FRF. Same comment for various species in Table 2 in last 2 columns

*done*

21, 2 Perhaps add 'the' between 'In' and 'case': *done*

[revised manuscript text omitted]

---

## Author Response (AR2)

We thank the reviewer and the editor for the additional comments, which we answer below.

**Editor comment:**

In addition (to the reviewer comment), please add an explanation of the non-SI unit "ppt" to the captions of Tables 3 to 5 and Figures 3 to 9 (or change the axis labels), so that they can be understood on their own, when taken out of context and without reference to the main text.

We have added the following sentence from the main text to the captions of the Tables and Figures mentioned:

*All values given here are mole fractions given in ppt, which is equivalent to pmol/mol.*

**Reviewer #2 comment:**

The change in emphasis from the recovery date of ozone to the new method for calculating EESC is a good one and the inclusion of model comparisons also add to the robustness of the results. The authors have generally addressed all of my previous comments so I support publication with consideration of the additional specific comments below.

Specific comments (page numbers refer to the combined document with changes tracked):

Page 31, line 12: change "chemical" to "chemically"  - *done*

Page 31, lines 18-25: I would be more specific when referring to the chemical lifetime here, and in general, as "local" or not. It's clear enough that you mean local chemical lifetime in lines 17-18 but then in line 23 it's not clear what you mean by "shorter lifetimes".

*Changed to 'local lifetimes'  (l. 17-18) and to 'shorter local lifetimes along the transport pathway' in line 23.*

Page 31, lines 27-28: "especially also of the" is awkward. I would suggest "…and on the path p it has taken, primarily the maximum path height (MPH), during transport." – *changed as suggested*

Page 43, line 19: change "which is" to "which was". "Output from this model run is available from the years 1960-2016." - *changed*

Page 43, line 20: "as a function" - *changed*

Page 43, line 20: Why are the model NH fractional release factors so much higher than observed? Is this significant?

*This is in principle not significant. It has been shown previously that the relation of inorganic chlorine to mean age (and thus fractional release) is very different from model to model (Waugh et al., 2007). The southern hemispheric values have been chosen, as the fractional release is much closer in TOMCAT here to observed fractional release in the Northern Hemisphere. By contrast the fractional release factors derived from EMAC are in very good agreement with observations.*

Page 44, line 14: "on a fixed mean age level" - *changed*

Page 44, line 16: "the year 2000" - *changed*

Page 44, line 17: change "was" to "were" - *changed*

Page 44, line 22: You should explain further what you mean and why you expect ozone mixing ratios to "follow" a pressure surface rather than a mean age surface in the future. Presumably the model output can verify this?

What we wanted to state here, is that EESC is a reasonable proxy for the influence of halogens on ozone in the future. E statement about ozone rather following a pressure level was awkward. We have therefore rephrased this as follows:

*The exact magnitude of changes in stratospheric dynamics is highly uncertain but it has been shown that ESC evaluated at pressure levels is a good proxy to describe the influence of halogens on the ozone column (Shepherd et al., 2014;Eyring et al., 2010). Based on the much better agreement of EESC with ESC at pressure levels, we conclude that EESC is a reasonable proxy for the effect of halogen loading on stratospheric ozone, given the overall high uncertainties associated to the future evolution of stratospheric dynamics.*

Page 45, line 9: replace "he" with "the" - *changed*

Page 64, line 5: "The model simulation shown here used prescribed trace gas scenarios, sea surface temperatures and sea ice content." - *changed*

Eyring, V., Cionni, I., Bodeker, G. E., Charlton-Perez, A. J., Kinnison, D. E., Scinocca, J. F., Waugh, D. W., Akiyoshi, H., Bekki, S., Chipperfield, M. P., Dameris, M., Dhomse, S., Frith, S. M., Garny, H., Gettelman, A., Kubin, A., Langematz, U., Mancini, E., Marchand, M., Nakamura, T., Oman, L. D., Pawson, S., Pitari, G., Plummer, D. A., Rozanov, E., Shepherd, T. G., Shibata, K., Tian, W., Braesicke, P., Hardiman, S. C., Lamarque, J. F., Morgenstern, O., Pyle, J. A., Smale, D., and Yamashita, Y.: Multi-model assessment of stratospheric ozone return dates and ozone recovery in CCMVal-2 models, Atmos. Chem. Phys., 10, 9451-9472, 10.5194/acp-10-9451-2010, 2010.
Shepherd, T. G., Plummer, D. A., Scinocca, J. F., Hegglin, M. I., Fioletov, V. E., Reader, M. C., Remsberg, E., von Clarmann, T., and Wang, H. J.: Reconciliation of halogen-induced ozone loss with the total-column ozone record, Nature Geosci, 7, 443-449, 10.1038/ngeo2155, 2014.
Waugh, D. W., Strahan, S. E., and Newman, P. A.: Sensitivity of stratospheric inorganic chlorine to differences in transport, Atmos. Chem. Phys., 7, 4935-4941, 10.5194/acp-7-4935-2007, 2007.

---

## Author Response (AR3)

We would like to thank the editor and the reviewers for the work they have invested into this manuscript. We agree that the quality has considerably improved during the review process.

**Reply to last editor comments**

Page 9/16 should read "species" - *done*

There are also various references to tables A1 and A2 - these should be changed to tables 1 and 2 before submitting the production files. - *changed*

Otherwise, I am happy to accept the paper for publication.

I would like to thank the 4 referees who all provided very thorough reviews and made significant and valuable contributions to the paper. I'd also like to commend the authors for their decision to include additional modelling results (and co-authors), which have greatly improved the scientific quality and relevance of this piece of work.